# Multi-Omics Analysis Reveals the Regulatory Mechanism of Probiotics on the Growth Performance of Fattening Sheep

**DOI:** 10.3390/ani14091285

**Published:** 2024-04-24

**Authors:** Mingyue Wang, Mingliang Yi, Lei Wang, Shixin Sun, Yinghui Ling, Zijun Zhang, Hongguo Cao

**Affiliations:** 1College of Animal Science and Technology, Anhui Agricultural University, Hefei 230036, China; mingyuewang@stu.ahau.edu.cn (M.W.); yimingliang@stu.ahau.edu.cn (M.Y.); yun249030715@163.com (L.W.); 2185028577@stu.ahau.edu.cn (S.S.); lingyinghui@ahau.edu.cn (Y.L.); zhangzijun@ahau.edu.cn (Z.Z.); 2Anhui Province Key Laboratory of Local Livestock and Poultry Genetic Resource Conservation and Bio-Breeding, Anhui Agricultural University, Hefei 230036, China

**Keywords:** probiotics, growth performance, microbial community, metabolism, transcriptome

## Abstract

**Simple Summary:**

Probiotics are beneficial bacteria that regulate the gut microbiota of animals, and are used to maintain the balance of the gut microbiota. However, there is limited research on probiotics as a feed ingredient for meat sheep. This study mainly explains the mechanism of the effect of adding probiotics to the diet on the growth performance of fattening sheep from six aspects: sequencing of rumen and fecal microorganisms; metabolomics of rumen, serum, and urine; and transcriptome sequencing of rumen epithelial cells. This study shows that adding probiotics to feed can improve the growth performance of fattening sheep. Our research results will provide a theoretical basis for better using probiotics in ruminant feeding.

**Abstract:**

Probiotics have been proven to improve the growth performance of livestock and poultry. The aim of this experiment was to investigate the effects of probiotic supplementation on the growth performance; rumen and intestinal microbiota; rumen fluid, serum, and urine metabolism; and rumen epithelial cell transcriptomics of fattening meat sheep. Twelve Hu sheep were selected and randomly divided into two groups. They were fed a basal diet (CON) or a basal diet supplemented with 1.5 × 10^8^ CFU/g probiotics (PRB). The results show that the average daily weight gain, and volatile fatty acid and serum antioxidant capacity concentrations of the PRB group were significantly higher than those of the CON group (*p* < 0.05). Compared to the CON group, the thickness of the rumen muscle layer in the PRB group was significantly decreased (*p* < 0.01); the thickness of the duodenal muscle layer in the fattening sheep was significantly reduced; and the length of the duodenal villi, the thickness of the cecal and rectal mucosal muscle layers, and the thickness of the cecal, colon, and rectal mucosal layers (*p* < 0.05) were significantly increased. At the genus level, the addition of probiotics altered the composition of the rumen and intestinal microbiota, significantly upregulating the relative abundance of Subdivision5_genera_incertae_sedis and Acinetobacter in the rumen microbiota, and significantly downregulating the relative abundance of *Butyrivibrio, Saccharofermentans*, and *Fibrobacter*. The relative abundance of *faecalicoccus* was significantly upregulated in the intestinal microbiota, while the relative abundance of *Coprococcus*, *Porphyromonas*, and *Anaerobacterium* were significantly downregulated (*p* < 0.05). There were significant differences in the rumen, serum, and urine metabolites between the PRB group and the CON group, with 188, 138, and 104 metabolites (*p* < 0.05), mainly affecting pathways such as vitamin B2, vitamin B3, vitamin B6, and a series of amino acid metabolisms. The differential genes in the transcriptome sequencing were mainly enriched in protein modification regulation (especially histone modification), immune function regulation, and energy metabolism. Therefore, adding probiotics improved the growth performance of fattening sheep by altering the rumen and intestinal microbiota; the rumen, serum, and urine metabolome; and the transcriptome.

## 1. Introduction

The emergence of antibiotics played an important role in animal growth and reproduction, disease prevention and treatment, etc. However, with the excessive abuse of antibiotics, bacteria in animals have developed resistance, disrupting the balance of animal gastrointestinal microbiota and, most importantly, it can also cause serious harm to human health [1,2]. China is a major country of the world’s livestock and poultry farming, as well as a major producer and user of antibiotics. In order to produce safe and high-quality livestock and poultry products and achieve healthy breeding, researchers urgently need to find ideal substitutes for antibiotic feed additives. Probiotics have the advantages of enhancing animal immunity and disease resistance, improving animal production performance and feed utilization, and are an ideal alternative to antibiotics [3,4,5]. The impact mechanism of probiotics on animals is mainly manifested through two aspects: regulating gastrointestinal health and improving the body’s immunity [6]. Probiotic additives can accelerate the growth and development of immune organs in ruminants by regulating the structure, quantity, and fermentation level of the gastrointestinal microbiota [7,8,9].

Animal probiotics, as one of the raw materials for microbial preparations, are currently mainly used in feed additives and silage [10,11]. Although probiotic products currently cannot completely replace antibiotic additives in diets, their application scope has gradually expanded and their reliability has improved since use began [12,13]. Probiotics have been shown to enhance the gut microbiota of livestock and poultry, and enhance immunity and growth performance [14,15,16]. At present, research on probiotics is very limited, mainly focusing on monogastrics, and there is still less research on ruminants, which to some extent limits the effective application of probiotics [17]. The development of multi-omics technology provides the possibility for a deeper understanding of the effects and mechanisms of probiotics.

The main probiotics used in ruminant feed include *Lactobacillus*, *Aspergillus oryzae*, *Streptococcus*, and *Bifidobacterium* [6,17,18]. Probiotic preparations mainly adjust the microbiota balance in an animal’s gastrointestinal tract, serving as the dominant microbiota in the animal gut to compensate for the number of normal microbiota, promote the formation of intestinal dominant microbiota, inhibit the growth and reproduction of pathogenic microorganisms, and restore the balance of gastrointestinal microbiota [19,20]. According to existing research findings, probiotics are defined as live microbial feed additives that beneficially affect host animals by enhancing intestinal balance, thereby improving feed utilization, nutrient absorption, growth rate, and economic benefits of poultry [21,22,23]. Preliminary experimental studies have found that the *Bacillus licheniformis* group (*Bacillus licheniformis*), *Bacillus subtilis* binary group (*Bacillus subtilis* and *Enterococcus faecalis*), *Bifidobacterium* triple group (*Bifidobacterium longum*, *Lactobacillus bulgaricus*, and *Streptococcus thermophilus*) and *Bifidobacterium* tetravalent group (*Bifidobacterium*, *Lactobacillus acidophilus*, *Enterococcus faecalis*, and *Bacillus cereus*) all improved the growth performance of fattening sheep, with the *Bifidobacterium tetravalent* group having the most significant effect (*p* < 0.05). We chose *Bifidobacterium tetravalent* for this experiment (Appendix A). Our research focuses on demonstrating the mechanism by which adding probiotics to the diet enhances the growth performance of fattening sheep. From multiple perspectives, including rumen microbiota, fecal microbiota, rumen metabolomics, serum metabolomics, urine metabolomics, and transcriptome sequencing of rumen epithelial cells, this will provide scientific guidance for the study of probiotics additives in ruminant feed.

## 2. Materials and Methods

### 2.1. Experimental Animals and Design

Twelve healthy Hu sheep (of similar age and weight) from Zhoushi Sheep Industry Co., Ltd. in Chuzhou, China, were selected for this experiment and randomly divided into two groups. The weight range of the fattening sheep in each group was 22.6 ± 1.9 kg. The control group (CON) was fed with a basic diet, and the formula and chemical composition are shown in Table 1. The experimental group (PRB) added probiotics (bifidobacteria quadruplex live bacteria: *Bifidobacterium*, *Lactobacillus acidophilus*, *Enterococcus faecalis*, and *Bacillus cereus*) to the control group’s diet, with a prefeeding period of 2 weeks and a formal feeding period of 2 months. During the experiment, each group was fed at a fixed time in the morning and evening, and the fattening sheep in each group fed and drank freely. Other treatments were carried out according to the daily feeding and management methods of the farm. The experimental plan was approved by the Animal Protection Committee of Anhui Agricultural University (NO: SYDW-P20190600601), and the experimental design and workflow are shown in Figure 1.

### 2.2. Test Samples and Data Collection

On the 1st and 60th day of the formal experiment, each group of experimental sheep was weighed, with an empty stomach before feeding, to calculate the average daily gain. The feed consumption of each group of fattening sheep during the 2-month experimental period was calculated to calculate the average daily feed intake. The feed conversion rate was calculated based on the average daily gain/average daily feed intake.

On the 60th day of the formal experiment, before feeding in the morning, the rumen fluid of each group was collected using a sheep stomach catheter. After filtering through 4 layers of gauze, it was quickly loaded into a 50 mL centrifuge tube, and centrifuged at 3000 r/min for 15 min. An amount of 1 mL was taken for analysis of the volatile fatty acids (VFAs), 2 mL was taken for a colorimetric analysis of the ammonia nitrogen content, and the remainder was stored at −80 °C for later use. An amount of 100 g of fresh fecal samples from each group of fattening sheep were collected, and store in liquid nitrogen for later use. Fasting blood was collected using a blood collection vessel without anticoagulants, and 10 mL of blood was collected from each Hu sheep and centrifuged at 3500 r/min for 15 min. The supernatant was used as the serum. The serum antioxidant indicators were analyzed using a reagent kit, including the total antioxidant capacity (T-AOC) (A015-1-2, Nanjing Institute of Biotechnology, Nanjing, China), superoxide dismutase (SOD) (A001-3-2, Nanjing Institute of Biotechnology, Nanjing, China), glutathione peroxidase (GSH-Px) (A005-1-2, Nanjing Jiancheng Institute of Biotechnology, Nanjing, China), and malondialdehyde (MDA) content (A003-1-2, Nanjing Jiancheng Institute of Biotechnology, Nanjing, China). The remaining was stored at −80 °C for future use. Fresh urine from the fattening sheep was collected and stored at −80 °C for future use. After the feeding experiment, all 12 fattening sheep in the CON and PRB groups were slaughtered (fasted for 12 h before slaughter), and the rumen epithelial tissue was collected and stored in liquid nitrogen for transcriptome sequencing. In addition, the intestinal tissue of the fattened sheep was collected, fixed with 4% paraformaldehyde solution, embedded in paraffin, and sliced. After HE staining, the morphological changes in each segment of intestinal tissue were observed.

### 2.3. Microbial Analysis of Rumen Fluid and Feces

Using a DNA extraction kit to extract microbial DNA from the rumen fluid and feces, 16S rRNA gene V3-V4 region universal primers were used to amplify the target sequence using a two-step PCR method. A Qubit 4.0 was used to measure the concentration of the extracted DNA and amplified sequence. Bioengineering (Shanghai, China) Co., Ltd. We used an Illumina MiSeq system for high-throughput sequencing, and analyzed the differences in microbial diversity and microbial community structure in the rumen and feces of fattening sheep between the CON and PRB groups.

### 2.4. Metabolomics Analysis of Rumen, Serum, and Urine

An amount of 10 μL of the previously preserved rumen fluid, serum, and urine samples were taken, and ultra-high-performance liquid chromatography–tandem quadrupole–time-of-flight mass spectrometry (UPLC-Q-TOF-MS) was used to perform a metabolomics analysis of the rumen fluid, serum, and urine of the fattening sheep. After processing and quality control of the obtained raw data, a multivariate statistical analysis was conducted; the conditions were that the *p*-value of Student’s *t*-test was less than 0.05 and the Variable Importance in the Projection (VIP) of the first principal component of the OPLS-DA model was greater than 1. Differential metabolites in the rumen fluid, serum, and urine were screened, and a metabolic pathway analysis was conducted to identify the key pathways with the highest correlation with the differential metabolites, and we analyzed the differences in the metabolites and metabolic pathways between the CON group and the PRB group of fattening sheep.

### 2.5. Transcriptomics Analysis

The RNA extraction and transcriptome sequencing of the rumen epithelial tissue samples were entrusted to Bioengineering (Shanghai, China) Co., Ltd. After RNA extraction, the samples were sequenced on the Illumina platform. After sequencing, data processing, and quality control, a differential expression analysis was performed using the expression level difference analysis software DESeq 2. Using a *p*-value < 0.05 and fold of difference |Fold Change | > 2 as the screening criteria, the differential expression of genes (DEGs) between the groups were screened. A comparative analysis was conducted between the DEGs and GO (Gene Ontology) databases to obtain the functional annotation information corresponding to the genes. Cluster Profiler was used for the functional enrichment analysis. When the *p*-value < 0.05, it was considered that the function was enriched, and the function of the differentially expressed genes was analyzed.

### 2.6. Data Analysis

The normal distribution of the data was tested using the SPSSAU Data Science Analysis Platform (https://spssau.com/, accessed on 6 February 2024). The Shapiro–Wilk test was used for normality testing. The experimental sample consisted of 6 replicates. In order to organize the data, Excel 2016 was used, and SPSS software (version 20.0, Chicago, IL, USA) was used to conduct independent sample t-tests on the data. *p* < 0.05 is the significant difference, and *p* < 0.01 is the extremely significant difference standard. The difference between the two levels is statistically significant, with *p* < 0.05 and *p* < 0.01, respectively.

## 3. Results

### 3.1. Growth and Performance

Before the formal experiment, we measured the initial weight of the CON group (6 fattening sheep) and the PRB group (6 fattening sheep). The average initial weight of the two groups was 23.40 ± 1.90 kg for the PRB group and 23.25 ± 1.54 kg for the CON group, respectively, with no significant difference (*p* > 0.05). After 2 months of formal feeding, the average daily gain of the PRB group of fattening sheep was 0.27 ± 0.02 kg, while that of the CON group was 0.21 ± 0.01 kg. The average daily weight gain of the PRB group of fattening sheep was significantly higher than that of the CON group (*p* < 0.05). The average daily feed intake was 1.20 ± 0.07 kg, with no significant difference (*p* > 0.05). The feed conversion rate of the PRB group was 0.225 ± 0.01, while that of the CON group was 0.170 ± 0.01. The feed conversion rate of PRB group of fattening sheep was significantly higher than that of the CON group (*p* < 0.05).

### 3.2. Morphological Characteristics of Gastrointestinal Tissue

As a ruminant, sheep have four stomachs, namely the rumen, reticulum, omasum, and abomasum. The rumen mainly has the function of storing and fermenting the feed, while the abomasum is the only digestive gland that secretes digestive enzymes and has true digestion. The rumen, reticulum, omasum, and abomasum of the fattening sheep were observed through tissue sectioning under an optical microscope, and the muscle layer thickness of the stomach was measured (Figure 2 and Table 2). Compared to the CON group, the thickness of the rumen muscle layer in the PRB group was significantly decreased (*p* < 0.01), while the changes in the thickness of the reticulum, omasum, and abomasum muscle layers were not significant (*p* > 0.05).

The intestines are the tissue that digests and absorbs nutrients and excretes feed residues in fattening sheep, including the small intestine and large intestine. The results show that adding probiotics to the diet significantly reduced the thickness of the duodenal muscle layer, increased the length of the duodenal villi, and significantly increased the thickness of the cecum and rectal mucosal muscle layer, as well as the thickness of the cecum, colon, and rectal mucosal layer in the fattening sheep (*p* < 0.05) (Figure 3 and Table 3).

### 3.3. Analysis of Rumen Fermentation Parameters and Rumen Microbiota

The rumen plays an important role in fermentation, and ammonia nitrogen in the rumen is the main nitrogen source for rumen microbial fermentation. We measured the fermentation parameters of the rumen (Table 4); the level of ammonia nitrogen in the PRB group was not significantly different from that in the CON group (*p* > 0.05). The total volatile fatty acid concentration in the PRB group was 88.69 mM, significantly higher than that in the CON group, at 66.10 mM (*p* < 0.01). The concentrations of acetate and propionate in the PRB group were 61.48 mM and 18.07 mM, respectively, significantly higher than those in the CON group, at 45.23 mM and 12.21 mM (*p* < 0.01). There was no significant difference in the concentrations of butyrate, valerate, and the acetate/propionate acid ratio between the PRB and CON groups (*p* > 0.05).

Based on the level of genus classification, the rumen microbial population was analyzed (Figure 4A). *Prevotella*, *Methanobrevibacter*, *Succiniclassicum*, *Clostridium IV*, *Selenomonas*, *Barnesiella,* and *Butyrivibrio* were detected in both the PRB and CON groups. *Prevotella*, *Methanobrevibacter*, and *Succiniclassicum* were the three dominant genera in the rumen of the fattening sheep. There were a total of five bacterial genera with significant differences between the PRB group and the CON group (*p* < 0.05) (Table 5). Compared to the CON group, the PRB group significantly upregulated two bacterial genera, namely Subdivision5_genera_incertae_sedis and Acinetobacter, and significantly downregulated three bacterial genera, namely *Butyrivibrio*, *Saccharofermentans*, and *Fibrobacter*.

### 3.4. Analysis of Fecal Microbiota

The community structure of the fecal microorganisms at the genus level was analyzed (Figure 4B), among which *Sporobacter*, *Bacteroides*, *Ruminococcus*, *Alistipes*, *Succinivibrio*, *Treponema,* and *Clostridium_XIVa* were detected in both the PRB and CON groups. *Sporobacter*, *Bacteroides*, and *Ruminococcus* were the three dominant genera of fecal bacteria detected in the PRB and CON groups. Four main microbial communities with significant differences between the CON and PRB groups were identified, namely *Coprococcus, Porphyromonas*, *Anaerobacterium*, and *Faecalicoccus* (Table 6). Compared to the CON group, the significantly upregulated genera in the PRB group were *Faecalicoccus*, while the significantly downregulated genera were *Coprococcus*, *Porphyromonas*, and *Anaerobacteria*.

### 3.5. Metabolomics Analysis of Rumen Fluid

Analyzing the rumen metabolite data of the 12 fattening sheep in the PRB and CON groups, a total of 188 significantly different metabolites were selected, including 60 anionic modes and 128 cationic modes (Appendix A). The distribution and change characteristics of the differential metabolites in the rumen of the two groups of fattening sheep were analyzed (Figure 5). It was observed that the differential metabolites, which mainly include organic acids and their derivatives, organic compounds, lipids and lipid-like molecules, as well as amino acids and their derivatives, are clearly distinguished from their metabolites in the rumen of the PRB and CON fattening sheep, with red indicating a relative upregulation of metabolites and blue indicating a relative downregulation of metabolites. According to the thermodynamic diagrams for anions and cations, these differential metabolites can be mainly divided into two categories: one has a total of 105 significantly downregulated differential metabolites compared to the CON group, such as L-Cystine, Coumestrol, Flutamide, N-tiglylglycine, and Diethyl sebacate. The other has a total of 73 significantly upregulated differential metabolites compared to the CON group, such as Sebacic acid, 2-hydroxy-3-methylbutyric acid, Jasmonic acid, Nicotinic acid, and Pyridoxine.

The metabolic pathway analysis of differential metabolites revealed a total of 26 metabolic pathways between the PRB and CON groups (Appendix A and Figure 6). It was found that vitamin B6 metabolism, Nicotinate and nicotinamide metabolism, taurine and hypotaurine metabolism, Riboflavin metabolism, and alpha-linolenic acid metabolism were the main metabolic pathways.

### 3.6. Serum Antioxidant Capacity and Metabolomics Analysis

The serum antioxidant level of fattening sheep is of great significance for the body’s ability to resist stress and adapt to the external environment. The antioxidant indicators GSH-Px, MDA, SOD, and T-AOC in the collected serum of the fattening sheep were tested. The test results are shown in Table 7. Compared with the CON group, the SOD (*p* < 0.01) and T-AOC (*p* < 0.05) in the serum of the fattening sheep in the PRB group were significantly increased. The GSH-Px in the serum of the fattening sheep in the PRB group was slightly higher than that of the CON group, and the MDA of the PRB group was slightly lower than that of the CON group (*p* > 0.05).

The analysis of the serum metabolite data revealed a total of 138 significantly different metabolites, including 42 anionic modes and 96 cationic modes (Appendix A and Figure 7). The differential metabolites were clearly distinguishable in the serum of the PRB and CON groups of fattening sheep, and were mainly divided into two categories: one had 108 significantly downregulated differential metabolites compared to the CON group, such as Eicosapentaenoic Acid, 5′-O-methylthymidine, Maltitol, L-Alanine, and Creatine; the other had a total of 30 significantly upregulated differential metabolites compared to the CON group, such as Citramalic acid, Pyrocatechol, vitamin E, Phenylethylamine, and 16-Hydroxypalmitic acid. 

A total of 38 serum metabolism-related pathways were analyzed between the PRB and CON groups (Appendix A and Figure 8) and Glycine, serine, and threonine metabolism; Terpenoid backbone biosynthesis; Arginine and proline metabolism; Phenylalanine metabolism; and beta-Alanine metabolism were the main metabolic pathways.

### 3.7. Urine Metabolomics Analysis

Urine, as the final excretion of metabolic products in fattening sheep, contains a large amount of metabolites and metabolic enzymes, which can reflect the biochemical metabolic status of fattening sheep; reveal changes in metabolic pathways, key metabolic pathways, and potential biomarkers; and have important research value. A total of 104 significantly different metabolites were screened, including 42 anionic modes and 62 cationic modes (Appendix A and Figure 9). The differential metabolites were mainly divided into two types: one type had 101 significantly downregulated differential metabolites compared to the CON group, such as Gentisic acid, L-Gulonic gamma-lactone, Sulfaphenazole, 5′-Deoxyadenosine, and Palmitic acid; the other type had a total of three significantly upregulated differential metabolites compared to the CON group, Cyanuric acid, D-Mannose, and 3.alpha.-Mannobiose.

We analyzed a total of 33 pathways related to urine metabolism between the PRB and CON groups (Appendix A and Figure 10), and found taurine and hypotaurine metabolism; Methane metabolism; Riboflavin metabolism; and Tryptophan metabolism; vitamin B6 metabolism; and Valine, leucine, and isoleucine biosynthesis were the main metabolic pathways.

### 3.8. Bioinformatics Analysis of circRNA Sequencing

Circular RNA (circRNA) is abundant in the cytoplasm of eukaryotic cells, mainly produced by pre-mRNA through variable splicing processing, and has many important regulatory functions [24]. It can block the inhibitory effect of miRNA on its target genes, regulate other types of RNA, and regulate protein activity [25]. circRNA is conserved in different species and has important research value. In order to screen for significantly different circRNAs, the screening conditions were set to *p*-value < 0.05 and the difference multiple |Fold Change| > 2, and the analysis results of significant expression differences in the circRNAs were obtained (Appendix A). We screened seven significantly differentially expressed circRNAs between the PRB and CON groups, of which three were significantly upregulated and four were significantly downregulated (Figure 11A).

For the screened differentially expressed circRNAs, GO enrichment analysis was used to study the distribution of the differentially expressed circRNA target genes using the annotation function (Figure 11B). These differentially expressed circRNAs mainly involve biological processes, such as participating in biological regulation, cellular processes, and metabolic processes. They are distributed in cellular locations, such as cells, cell components, and organelles, and participate in both binding and catalytic activity molecular functions. We used enrichment analysis to examine the biological functions of the differentially expressed circRNA target genes and identified the top-30 differentially expressed circRNA target genes with the highest degree of functional enrichment (Table 8 and Figure 11C). These circRNA target genes were mainly involved in protein modification regulation, especially histone modification and immune response expression and regulation.

### 3.9. Bioinformatics Analysis of lncRNA Sequencing

Long-stranded noncoding RNAs (lncRNAs) are a type of noncoding RNA with a length exceeding 200 nt that regulate gene expression at the epigenetic, transcriptional, and post-transcriptional levels by binding to DNA, RNA, or proteins [26,27,28]. A differential expression analysis was conducted on RNA transcripts, and differential expression transcripts between the PRB and CON groups were obtained (Appendix A). A total of 20 significantly differential transcripts were screened, including 6 significantly upregulated lncRNAs and 3 mRNAs, and 2 significantly downregulated lncRNAs and 9 mRNAs (Figure 12A).

A differential expression functional annotation analysis was conducted on the screened differentially expressed transcripts. The corresponding gene functions of the differentially expressed transcripts were mainly concentrated in the cellular processes, biological regulatory processes, and metabolic processes involved in biological processes; distributed in positions such as cells, cell components, and organelles; and involved in binding and catalytic active molecular functions. This is consistent with the distribution of the differentially expressed circRNA target genes using the annotation functions through a GO enrichment analysis (Figure 12B). The gene enrichment analysis identified the top-30 functions with the highest enrichment levels (Table 9 and Figure 12C). The genes corresponding to the differential transcripts of the fattening sheep fed with a probiotic diet were mainly enriched in regulating immune cell functions, such as natural killer cells and white blood cell regulation. This is consistent with probiotics increasing the number and activity of immune cells, such as increasing the number of white blood cells and natural killer cells, and improving the activity and survival ability of immune cells.

### 3.10. miRNA Sequencing Bioinformatics Analysis

MicroRNAs (miRNAs) are a type of RNA that plays an important regulatory role, and mRNA expression is regulated in multiple ways, among which miRNAs are one of the key regulators. miRNA is involved in various regulatory pathways, including development, viral defense, hematopoietic processes, organ formation, cell proliferation and apoptosis, and fat metabolism [29,30]. After a differential expression analysis, a total of 1067 differentially expressed miRNAs were screened, of which 626 were significantly upregulated and 441 were significantly downregulated (Figure 13A).

The functional annotation classification of the target genes for the differential miRNA targeting mRNA (Figure 13B) shows that the target genes for the differential miRNA targeting mRNA are mainly concentrated in biological processes, such as cellular processes and biological regulatory processes; are distributed in positions such as cells, organelles, and membranes; and play a role in binding and catalytic activity molecules. This was similar to the GO enrichment analysis of the distribution of circRNA and lncRNA using the annotation function. We screened the top-10 functional analysis results with the highest enrichments in each of the three ontologies (Table 10 and Figure 13C). In the fattening sheep fed with a probiotic diet, the target genes for the miRNA targeting mRNA were mainly enriched in transcriptional regulation and protein phosphorylation modification, which play an important regulatory role in protein, ATP, and nucleoside binding, and play an important role in animal growth and development.

## 4. Discussion

Research has shown that probiotics can play an important role as the best alternative to antibiotics, mainly through the interaction of bacteria–bacteria and host–bacteria [10]. They have many beneficial aspects, such as anti-oxidation, anti-inflammatory, anti-allergy, anti-cancer, anti-mutation, anti-diabetes, and anti-virus potential, and they can also be used as a growth promoter for animals [31,32,33,34]. Probiotics are defined as live microbial feed additives that beneficially affect host animals by enhancing intestinal balance, improving feed efficiency, nutrient absorption, and growth rate [35]. Research has found that probiotics can be used as useful bacteria with which to eliminate pathogenic bacteria, and their potential applications can help improve the growth performance of livestock and poultry, improve the digestion and absorption capacity of nutrients, promote production, and ensure the health of livestock [36,37,38]. In this study, compared with the CON group, the PRB group was fed 1.5 × 10^8^ CFU/g probiotics daily, and the average daily weight gain of the PRB group was significantly higher than that of the CON group (*p* < 0.05), indicating that the addition of probiotics significantly improved the growth performance of the fattening sheep, which is consistent with previous research results showing that probiotics improve livestock and poultry growth performance.

The main characteristic of ruminants is the fermentation of microorganisms in the rumen, and the rumen volume accounts for 80% of the entire stomach volume [39]. There are a large number of ciliates and bacteria in the rumen, which play an important role in the rumen fermentation process. Research has found that probiotics can regulate gastrointestinal microbiota and promote digestion [40,41]. Feeding probiotics improves the composition of the rumen microbiota in fattening sheep, enhances microbial fermentation, and facilitates the absorption and utilization of cellulose and hemicellulose into volatile fatty acids, promoting protein decomposition and synthesis [42]. The rumen muscle layer thickness of the PRB fattening sheep was significantly lower than that of the CON (*p* < 0.01), due to the enhanced digestion of microorganisms in the rumen, which does not require strong peristalsis of food in the rumen, resulting in a thinning of the rumen muscle layer thickness. At the same time, it reduces the energy consumption of rumen peristalsis, and this experiment confirms that supplementing with probiotics improves the composition of rumen microorganisms and improves the digestion efficiency of fattening sheep feed. 

The intestine is an important site for the digestion and absorption of nutrients, and the efficient digestion and absorption of feed nutrients plays an important role in the healthy growth of livestock [43]. The intestinal villi on the surface of the duodenal intima increase the surface area of the intestine, allowing for more nutrients to be absorbed. In addition, the cells on the surface of the intestinal villi secrete more digestive enzymes, enabling efficient digestion and the absorption of feed nutrients [44]. In this experiment, the length of the duodenal villi in the PRB group was significantly higher than that in the CON group (*p* < 0.05), indicating that adding probiotics helps improve the digestion and absorption of nutrients in fattening sheep. The peristalsis of the intestine mainly relies on the contraction ability of the muscular layer. The developed muscular layer can promote intestinal peristalsis and enhance the absorption and utilization of nutrients. The PRB group had no significant difference except for the mucosal muscular layer of the colon (*p* > 0.05), while the mucosal layer and muscular layer of the cecum, colon, and rectum significantly increased, confirming that the PRB group were more efficiency at nutrient absorption and utilization (*p* < 0.05).

In this study, supplementing with probiotics significantly increased the total volatile fatty acid concentration (*p* < 0.01), indicating that probiotics have an improved effect on rumen microbial fermentation. The increase in total volatile fatty acid level indicates a high energy supply level for fattening sheep, and the average daily weight gain of the PRB group of fattening sheep significantly increased. The gastrointestinal microbiota is a signaling hub that links the host’s metabolism and immune system. Adding probiotics to the diet helps to establish and maintain suitable microbiota in the gastrointestinal tract [45].

A rumen microbiota analysis revealed a significant difference between the PRB group and the CON group in their total of five bacterial genera (*p* < 0.05). Compared to the CON group, the PRB group significantly upregulated Subdivision5_genera_incertae_sedis and Acinetobacter, and significantly downregulated *Butyrivibrio*, *Saccharofermentans*, and *Fibrobacter*. The rumen, as a unique digestive organ of ruminants, plays an important role. Improving the composition of rumen microbiota by supplementing with probiotics is of great significance for improving the growth performance of fattening sheep [46]. Probiotics have been shown to have various beneficial physiological effects on animal intestines. An analysis of fecal microorganisms revealed that the significantly upregulated genus in the PRB group was *Faecalicoccus*, and the significantly downregulated genera included *Coprococcus, Porphyromonas*, and *Anaerobacterium*. Supplementing with probiotics in the diet is beneficial for promoting the reproduction of intestinal microbiota, maintaining the balance of intestinal microbiota, increasing intestinal patency, enhancing intestinal protective function, regulating the body’s metabolism, promoting the growth and development of fattening sheep, and enhancing immunity.

To further explore the mechanism of adding probiotics to the diet to enhance the growth performance of fattening sheep, we conducted a metabolomics analysis on the rumen fluid, serum, and urine. Significant changes in metabolites were observed in the rumen fluid, serum, and urine of fattening sheep supplemented with probiotics. The pyruvate in serum differential metabolites is mainly involved in the entire basic metabolic process through the acetyl-CoA and tricarboxylic acid cycle pathways, including the conversion between sugars, fats, and amino acids [47]. These differential metabolites play an important role in energy metabolism and amino acid metabolism.

An analysis of the key metabolic pathways found that supplementing with probiotics had an impact on taurine and taurine metabolic pathways, vitamin B6 and vitamin B2, and a series of amino acid metabolism pathways. Taurine is a widely present free amino acid that participates in many different physiological processes, such as bile acid binding, osmotic regulation, antioxidant activity, calcium signaling regulation, and liver detoxification. It is the main biosynthetic compound of the liver and can improve gastric injury through its antioxidant properties [48,49]. Vitamin B6 plays a role in the various metabolic functions of proteins, carbohydrates, and lipids. At the same time, it can also participate in the conversion of tryptophan into niacin or 5-hydroxytryptamine, which can enhance immunity [50]. Vitamin B2 is involved in the tricarboxylic acid cycle and the catalytic action of multiple enzymes in the respiratory chain, promoting glucose, fat, and protein metabolism, thereby generating energy. Its metabolites FAD and FMN can serve as coenzymes that participate in electron transfer and ATP synthesis in the mitochondrial respiratory chain, maintaining the normal respiratory function of cells. It can also serve as an antioxidant, protecting cells from free radical damage and slowing down the aging process [51]. The results of the metabolomics analysis suggest that probiotics can play a role in promoting growth and development by regulating energy and amino acid metabolism pathways.

In addition, we screened differential genes using a GO functional enrichment analysis. In the PRB and CON groups, we screened target genes with a significant differential expression of circRNA, lncRNA, mRNA, and miRNA, which are mainly related to cellular processes, biological regulation, and metabolic processes. They are mainly enriched in protein modification regulation (especially histone modification), immune function regulation, and energy metabolism pathways, which play important roles in animal growth and development.

Previous studies have found that adding probiotics to livestock and poultry feed can improve the digestive system’s beneficial microbiota, increase the feed digestion and absorption rate, and regulate the body’s immune system [52,53,54]. These studies are consistent with the mechanism of adding probiotics to improve the growth performance of fattening sheep in this experiment, supporting the finding that supplementing with probiotics can improve the growth performance of fattening sheep. In the future, we will conduct in-depth research on the mechanisms of probiotics in healthy aquaculture, laying a solid theoretical foundation for the widespread application of probiotics in the field of animal husbandry.

## 5. Conclusions

In summary, probiotics improve the gastrointestinal microbiota and nutritional metabolism of fattening sheep, improve the immunity and digestive and absorption functions of fattening sheep, and have a positive effect on improving the growth performance of fattening sheep. It can play the same role as antibiotics, which provides a theoretical basis for the healthy breeding of fattening sheep.

## Figures and Tables

**Figure 1 animals-14-01285-f001:**
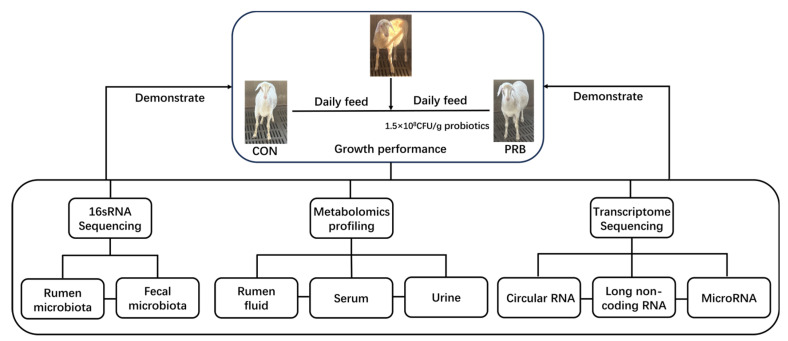
Experimental design and workflow for feeding fattening sheep with probiotics diet, including rumen and fecal microbiome, rumen fluid, serum and urine metabolome, and transcriptome sequencing of rumen epithelial cells. Twelve Hu sheep were randomly assigned to a basal diet (CON) or a basal diet supplemented with 1.5 × 10^8^ CFU/g probiotics (PRB).

**Figure 2 animals-14-01285-f002:**
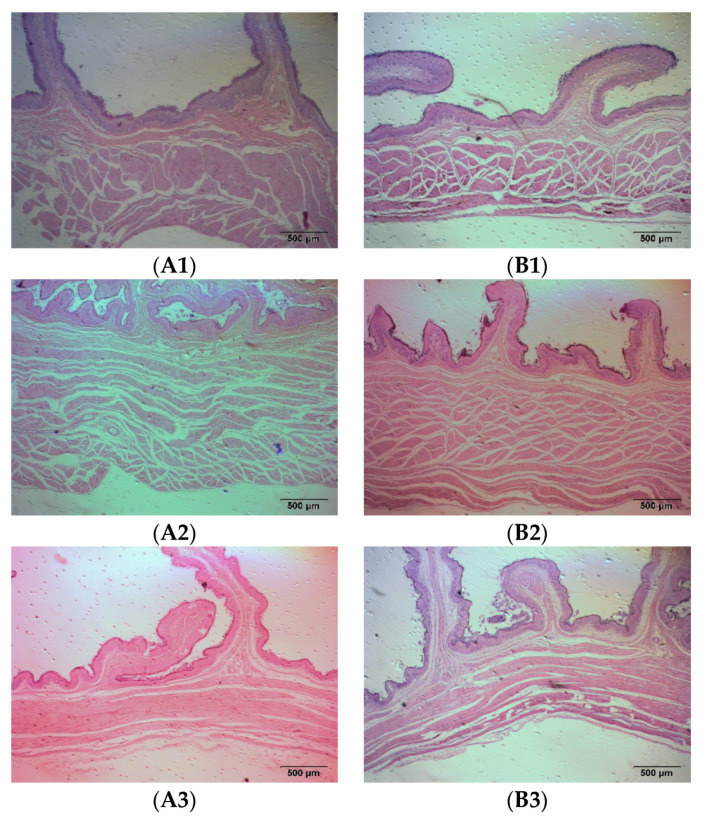
Histomorphology of stomachs of fattening sheep fed with PRB diet. (**A**) CON, (**B**) PRB; (1) rumen, (2) reticulum, (3) omasum, (4) abomasum.

**Figure 3 animals-14-01285-f003:**
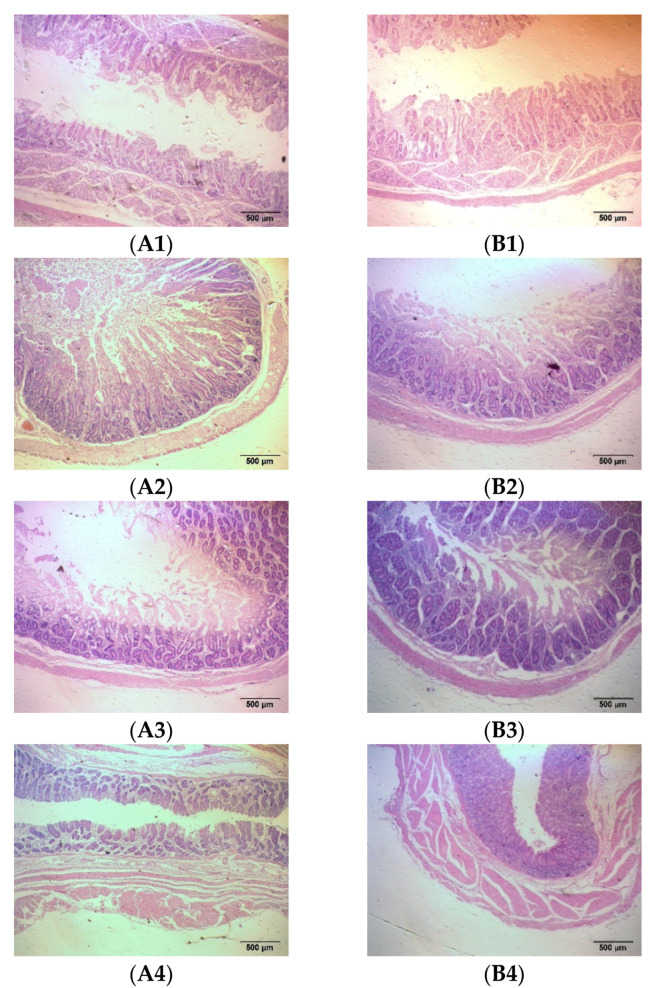
Histomorphology of intestinal segments of fattening sheep fed with PRB diet. (**A**) CON, (**B**) PRB; (1) duodenum, (2) jejunum, (3) ileum, (4) cecum, (5) colon, (6) rectum.

**Figure 4 animals-14-01285-f004:**
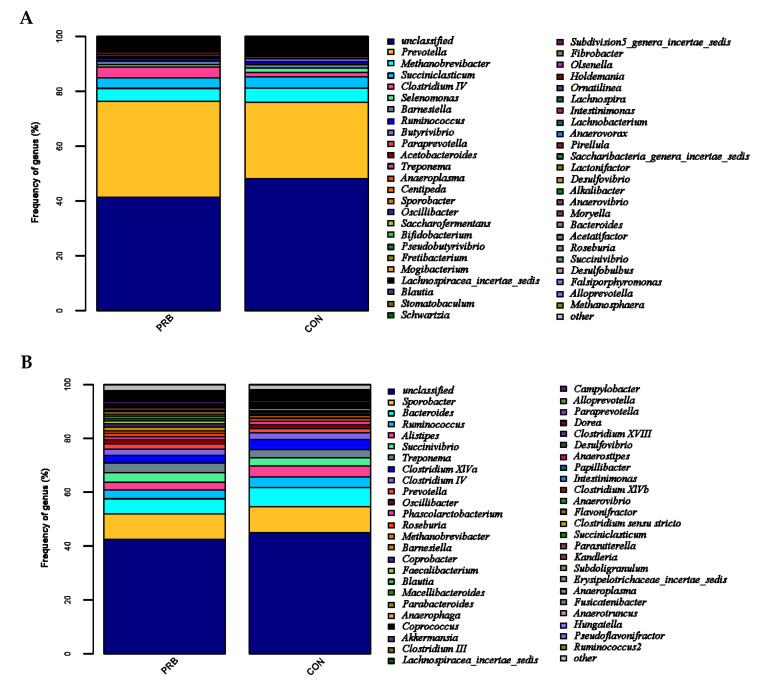
Distribution characteristics of horizontal microbiota in the PRB group. (**A**) Distribution of rumen microbiota and (**B**) distribution of fecal microbiota.

**Figure 5 animals-14-01285-f005:**
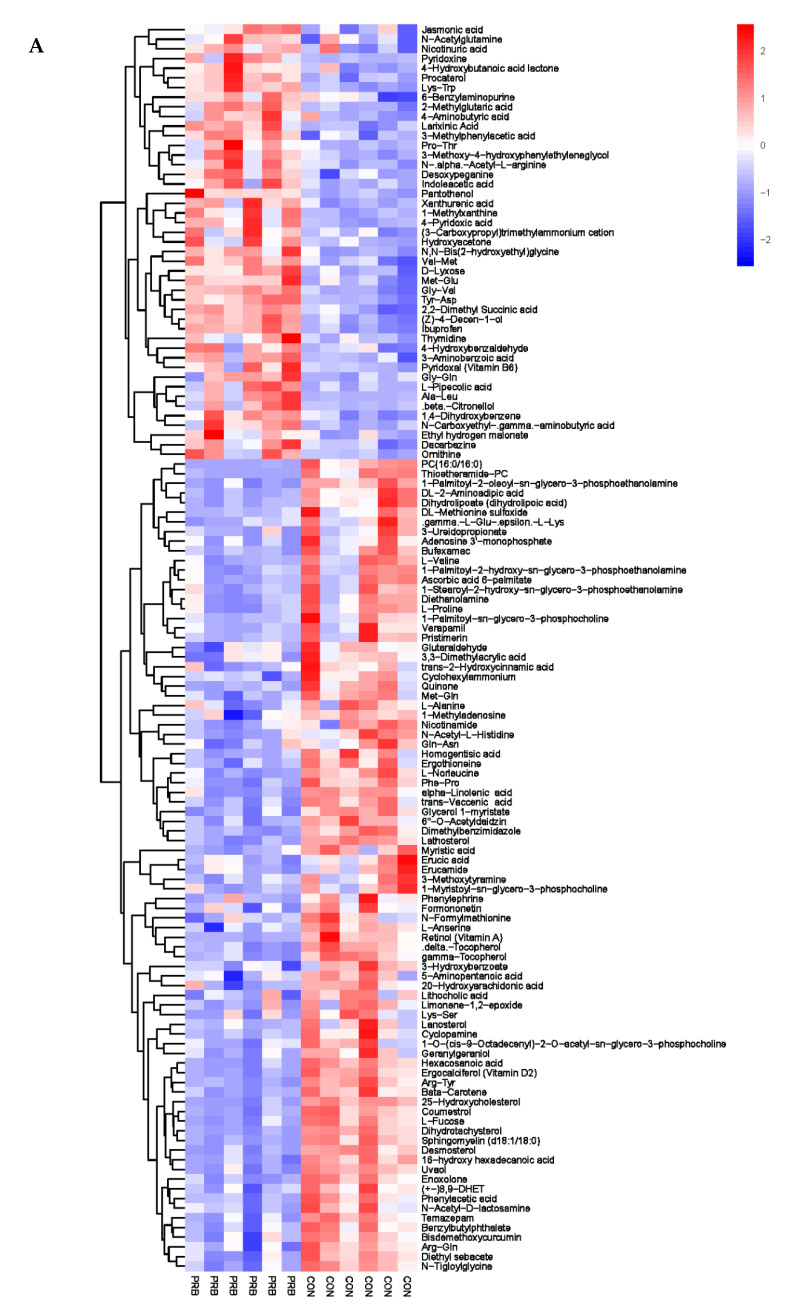
Hierarchical cluster analysis of rumen metabolites in fattening sheep. (**A**) Hierarchical clustering analysis of PRB group in cationic mode, (**B**) hierarchical cluster analysis of PRB group in anionic mode. Note: The horizontal axis represents the different experimental groups, the vertical axis represents the differential metabolites compared in the group, the blue color block represents that the relative expression level of the corresponding position metabolites is downregulated, and the red color block represents upregulation.

**Figure 6 animals-14-01285-f006:**
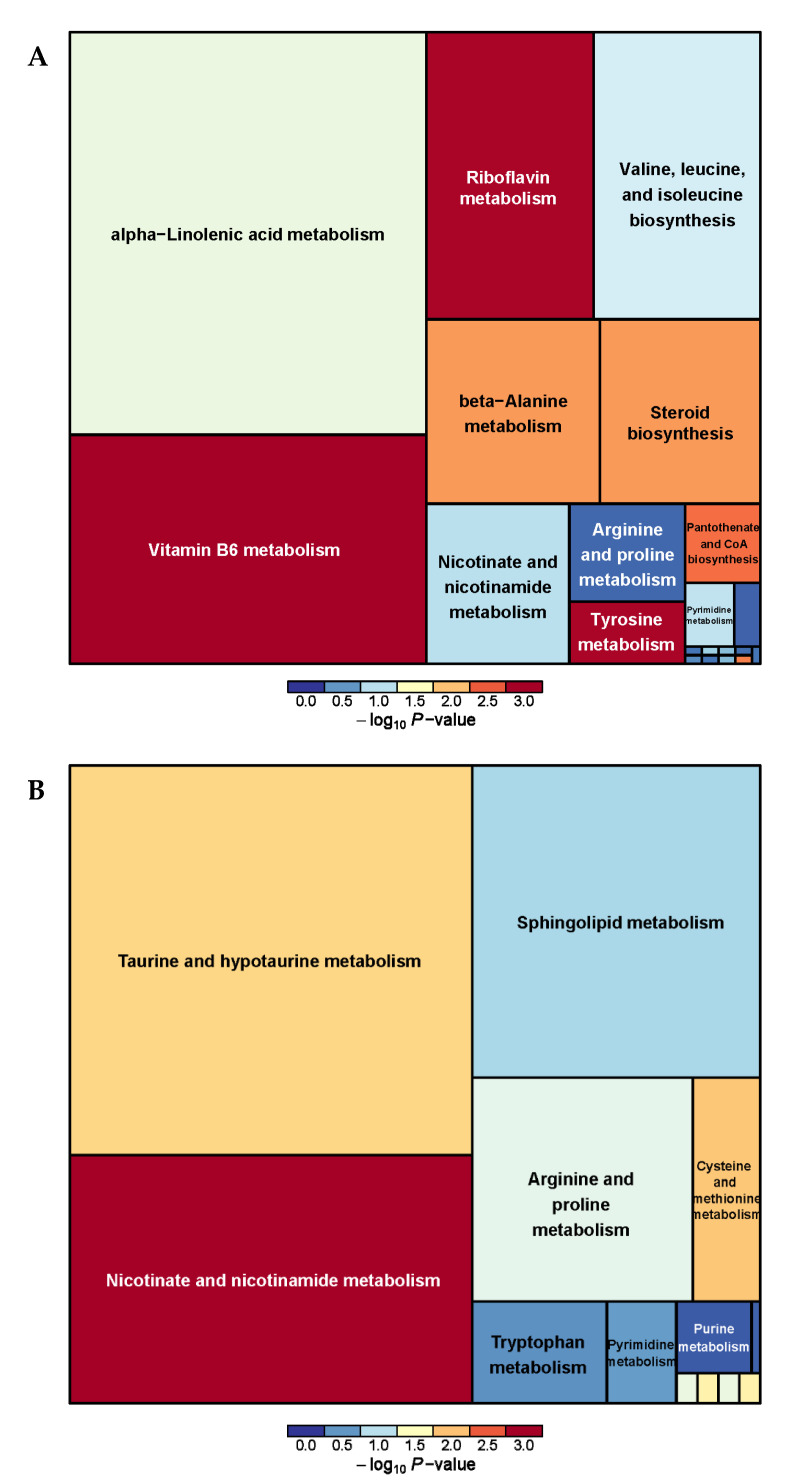
Pathway analysis of rumen metabolomics in fattening sheep. (**A**) Pathway analysis of PRB group in cationic mode, (**B**) pathway analysis of PRB group in anionic mode. Note: The color and size of the squares indicate the impact of mint treatment on sample metabolism, while larger red squares indicate a greater impact on the pathway.

**Figure 7 animals-14-01285-f007:**
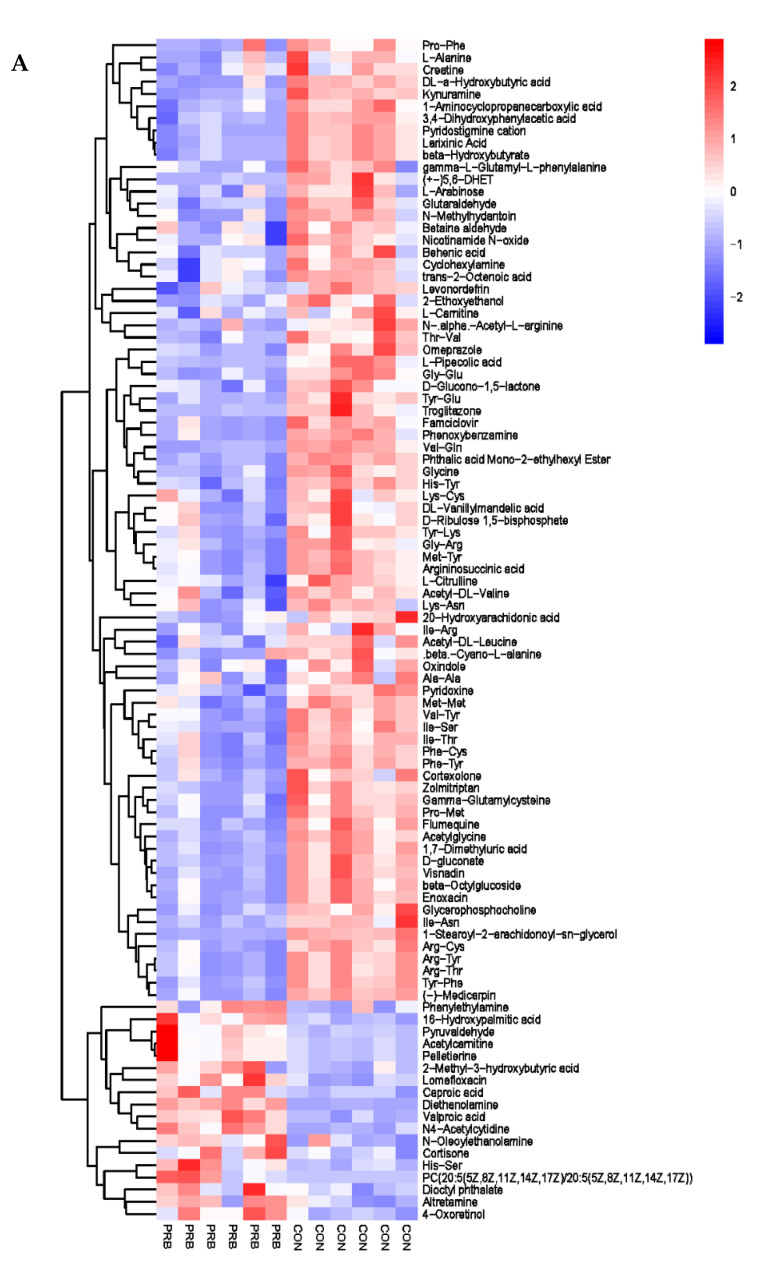
Hierarchical cluster analysis of serum metabolites in fattening sheep. (**A**) Hierarchical cluster analysis of PRB group serum in cationic mode, (**B**) hierarchical cluster analysis of PRB group serum in anionic mode.

**Figure 8 animals-14-01285-f008:**
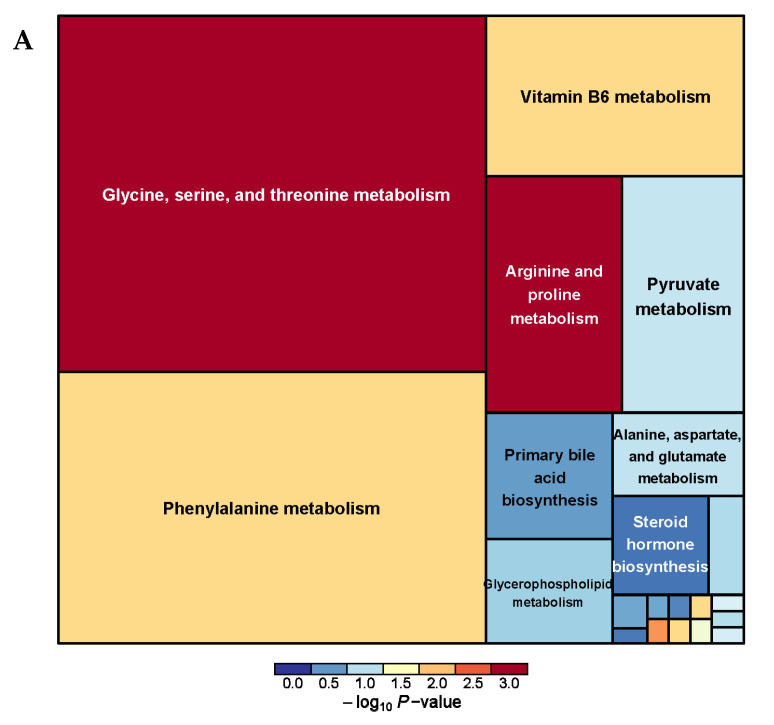
Pathway analysis of serum metabolomics. (**A**) Pathway analysis of PRB group serum in cationic mode, (**B**) pathway analysis of PRB group serum in anionic mode.

**Figure 9 animals-14-01285-f009:**
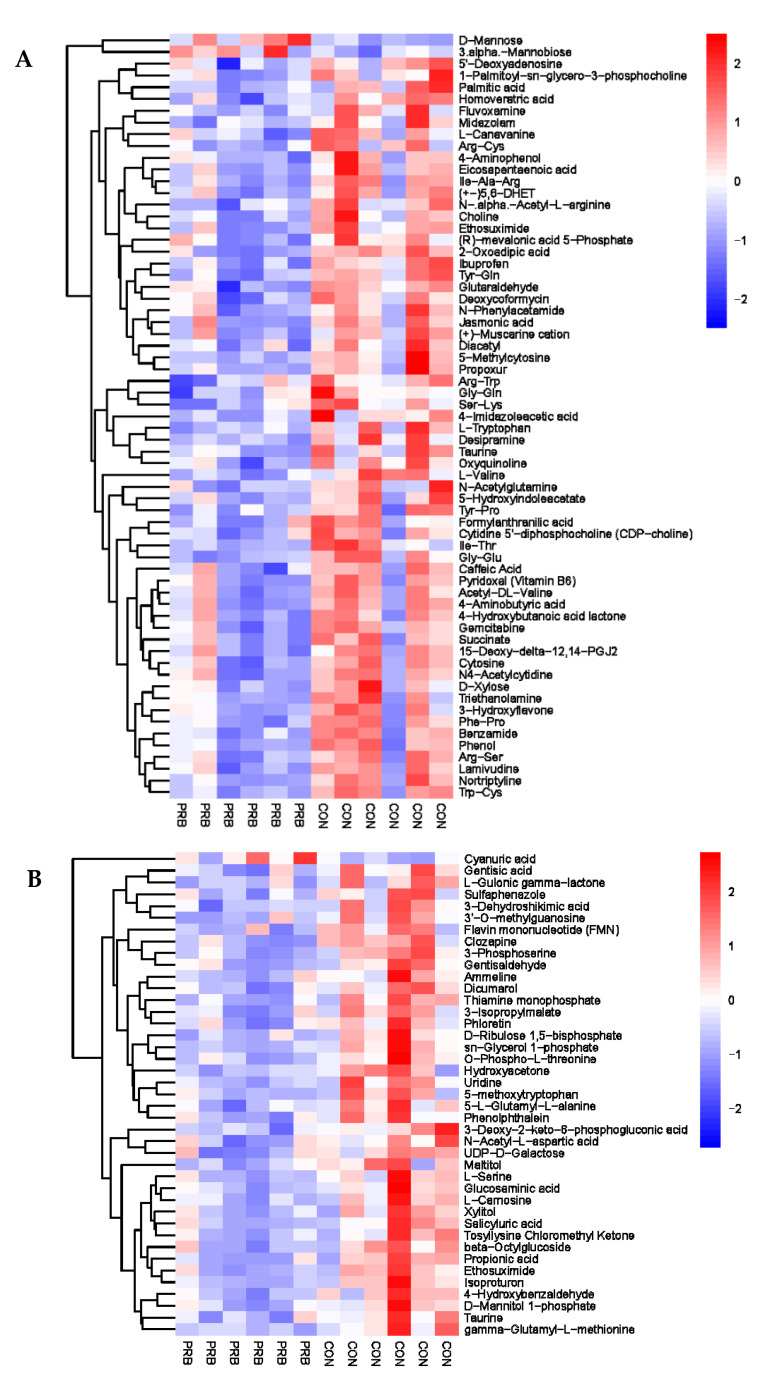
Hierarchical cluster analysis of urine metabolites in fattening sheep. (**A**) Hierarchical cluster analysis of PRB group urine in cationic mode, (**B**) hierarchical cluster analysis of PRB group urine in anionic mode.

**Figure 10 animals-14-01285-f010:**
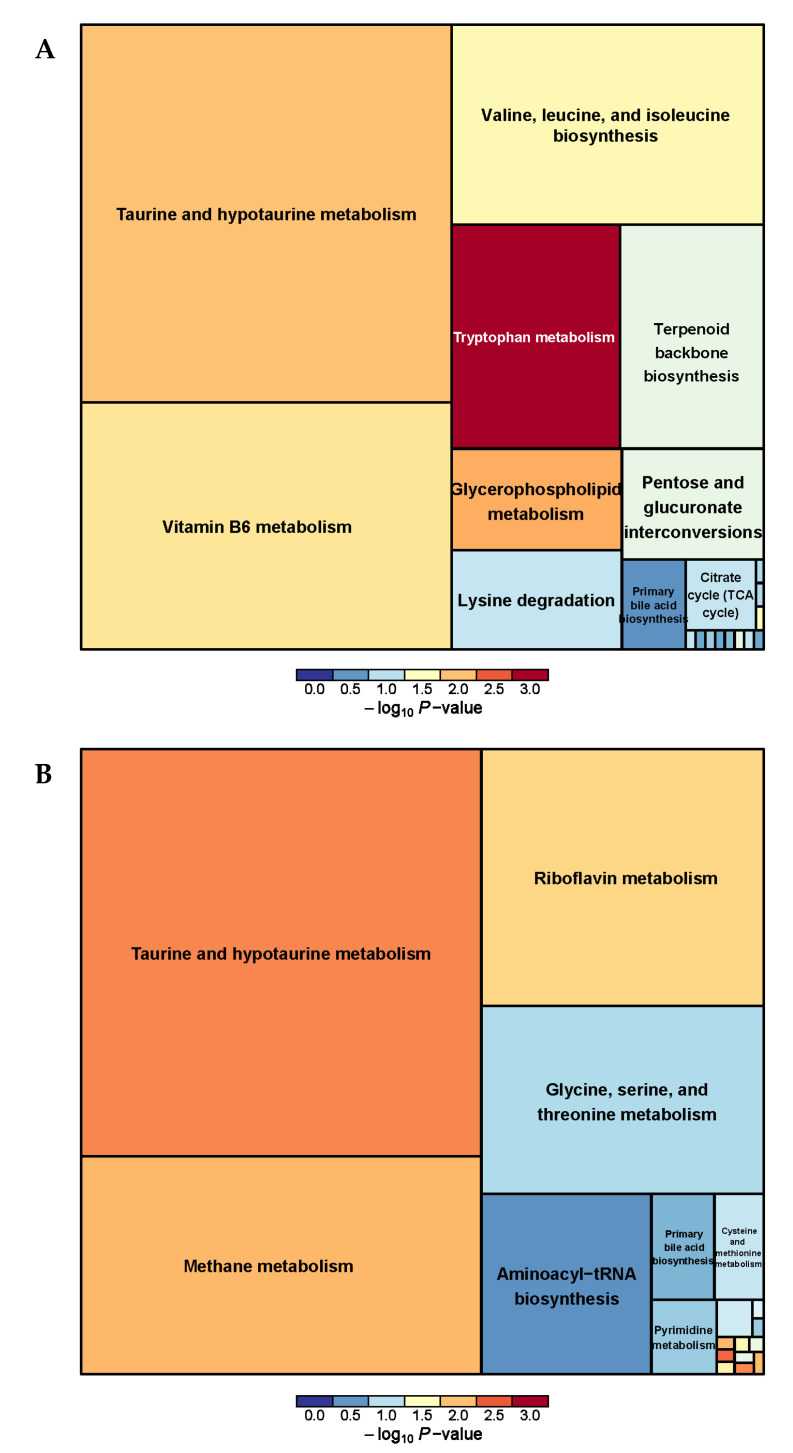
Pathway analysis of urine metabolomics. (**A**) Pathway analysis of PRB group urine in cationic mode, (**B**) pathway analysis of PRB group urine in anionic mode.

**Figure 11 animals-14-01285-f011:**
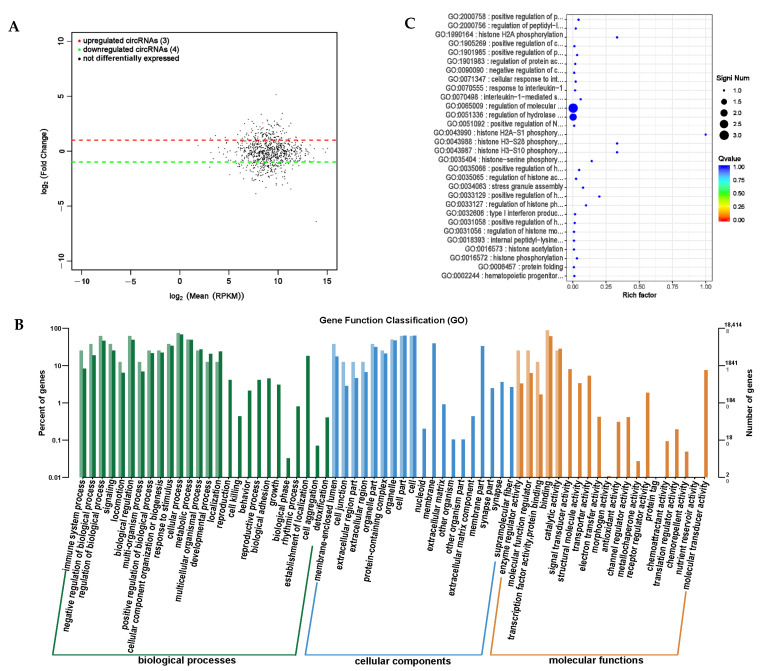
Differential expression analysis of circRNA between PRB group and CON group. (**A**) circRNA expression differential MA map. The horizontal axis represents the mean of log_2_ (RPKM) between the samples, with log_2_ (**A**) + log_2_ (**B**))/2, and the vertical axis represents the value of log_2_ (Fold Change), which is the value of log_2_ (**B**,**A**). Each point on the graph represents a transcript, with red indicating upregulation, green indicating downregulation, and black indicating nondifferential transcripts. (**B**) Differential expression of circRNA host gene functional annotation classification histogram. The horizontal axis represents the functional classification, while the vertical axis represents the number of genes within the classification (**right**) and their percentage in the total number of annotated genes (**left**). Different colors represent different classifications; light colors represent the host genes, and dark colors represent all genes. (**C**) A scatter plot of the top-30 functions with significant enrichment of differential circRNA. The vertical axis represents functional annotation information, while the horizontal axis represents the Rich factor corresponding to the function. The size of the Qvalue is represented by the color of the dot. The smaller the Qvalue, the closer the color is to red. The number of differentially expressed circRNA host genes included in each function is represented by the size of the dot.

**Figure 12 animals-14-01285-f012:**
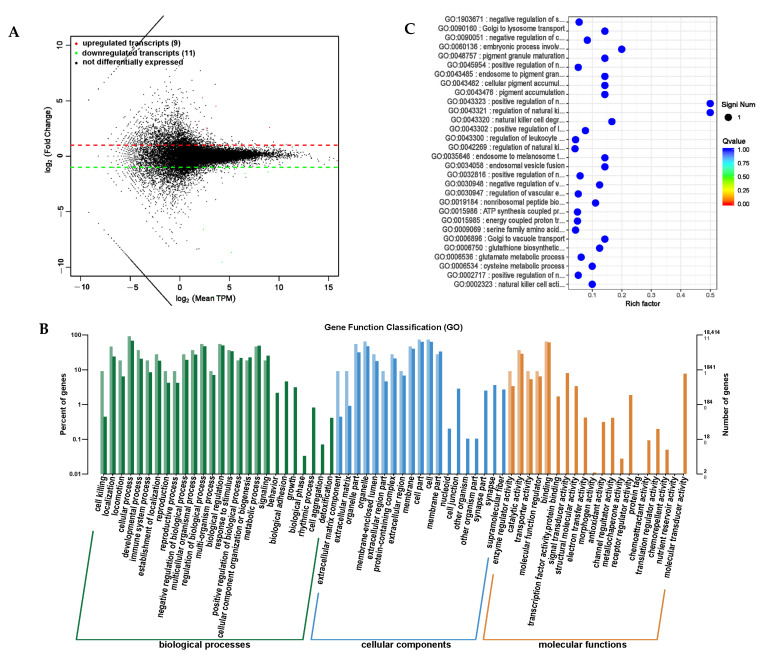
Differential expression analysis of transcriptome (lncRNA, mRNA) between PRB group and CON group. (**A**) Transcript expression differential MA map. (**B**) Differential transcript GO annotation classification bar chart. (**C**) Scatter plot of top-30 functions with significant enrichment of differential transcripts.

**Figure 13 animals-14-01285-f013:**
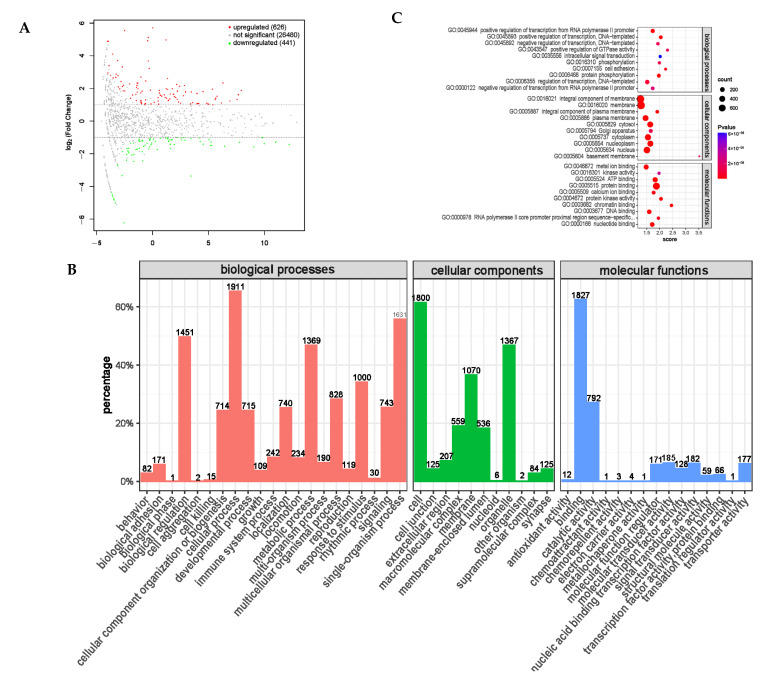
Differential expression analysis of miRNA between PRB group and CON group. (**A**) miRNA expression differential MA map. (**B**) Differential expression miRNA GO annotation classification histogram. (**C**) Top 10 × 3 (biological process, cellular component, and molecular function, top 10 for each) functional scatter plot shows significant enrichment of differential miRNAs.

**Table 1 animals-14-01285-t001:** Main components of the diet for fattening sheep during the experimental period.

Item	CON	PRB
Ingredient (%)		
Ground corn grain	28.00	28.00
Soybean meal, 43.5% CP	15.00	15.00
Rapeseed meal, 36.3% CP	9.00	9.00
Wheat bran	4.00	4.00
Sodium bicarbonate	1.00	1.00
Salt	1.00	1.00
Dicalcium phosphate	0.50	0.50
Calcium carbonate	0.50	0.50
Premix ^1^	1.00	1.00
Peanut straw	15.00	15.00
Soybean straw	25.00	25.00
Probiotics (CFU/g)	0.00	1.5 × 10^8^
Chemical composition (%)		
CP	15.10	15.10
NDF	38.70	38.70
ADF	23.20	23.20
Ether extract	3.10	3.10
Calcium	0.75	0.75
Total phosphorus	0.43	0.43
Metabolizable energy ^2^, MJ/kg	9.83	9.83

Abbreviations: CON = control diet; PRB = *Probiotic Bifidobacterium tetraureticum*. ^1^ Formulated to provide (per kilogram of premix) 600,000 IU of vitamin A, 80,000 IU of vitamin D3, 5000 IU of vitamin E, 8000 mg of Zn, 60 mg of Se, 200 mg of I, 9400 mg of Fe, 72 mg of Co, 10,400 mg of Mn, and 1600 mg of Cu. ^2^ Calculated according to Ministry of Agriculture of P.R. China, 2004.

**Table 2 animals-14-01285-t002:** The measurement of gastric tissue thickness.

Item	Treatment		SEM	*p*-Value
	CON	PRB		
Rumen, μm	1010.62 ^A^	762.14 ^B^	20.277	0.000
Reticulum, μm	1335.28	1214.42	48.655	0.072
Omasum, μm	706.78	751.04	33.285	0.232
Abomasum, μm	451.64	411.71	14.340	0.060

Note: Means in the same row with different superscripts ^A^, ^B^ are significantly different, with *p* < 0.01.

**Table 3 animals-14-01285-t003:** The measurement of intestine tissue thickness.

Position	Item	Treatment		SEM	*p*-Value
		CON	PRB		
Duodenum, μm	Muscularis	431.17 ^a^	335.07 ^b^	15.235	0.032
	Villus	383.11 ^b^	497.72 ^a^	14.747	0.041
Jejunum, μm	Muscularis	272.98	253.20	10.134	0.321
	Villus	362.08	303.83	19.871	0.063
Ileum, μm	Muscularis	161.41	191.14	7.468	0.058
	Villus	276.27	292.31	6.887	0.071
Cecum, μm	Muscularis mucosae	344.66 ^b^	507.54 ^a^	10.476	0.036
	Mucosa	396.35 ^b^	565.47 ^a^	13.394	0.041
Colon, μm	Muscularis mucosae	276.68	241.48	6.477	0.078
	Mucosa	535.91 ^b^	690.99 ^a^	19.242	0.021
Rectum, μm	Muscularis mucosae	363.97 ^b^	496.48 ^a^	6.081	0.036
	Mucosa	351.72 ^b^	490.60 ^a^	17.761	0.019

Note: Means in the same row with different superscripts ^a^, ^b^ are significantly different, with *p* < 0.05.

**Table 4 animals-14-01285-t004:** Rumen fermentation characteristics of probiotics diet treatment.

Item	Treatment		SEM	*p*-Value
	CON	PRB		
Ammonia nitrogen, mg/dL	6.21	8.52	1.483	0.140
Total volatile fatty acid, mM	66.10 ^B^	88.69 ^A^	4.055	0.000
Acetate, mM	45.23 ^B^	61.48 ^A^	3.258	0.000
Propionate, mM	12.21 ^B^	18.07 ^A^	1.090	0.000
Butyrate, mM	7.85	8.33	0.345	0.187
Valerate, mM	0.81	0.80	0.065	0.909
Acetate: Propionate	3.71	3.45	0.217	0.250

Note: Means in the same row with different superscripts ^A^, ^B^ are significantly different, with *p* < 0.01.

**Table 5 animals-14-01285-t005:** Significant changes in the main microbial communities in the rumen of the PRB diet group.

Phylum	Genus	Treatment		SEM	*p*-Value
		CON	PRB		
*Firmicutes*	*Butyrivibrio*	1.095 ^A^	0.522 ^B^	0.175	0.005
*Bacteroidetes*	*Saccharofermentans*	0.325 ^a^	0.183 ^b^	0.066	0.049
*Verrucomicrobia*	Subdivision5_genera_incertae_sedis	0.065 ^b^	0.190 ^a^	0.045	0.013
*Fibrobacteres*	*Fibrobacter*	0.222 ^a^	0.048 ^b^	0.075	0.036
*Proteobacteria*	Acinetobacter	0.023 ^b^	0.035 ^a^	0.005	0.031

Note: Means in the same row with different superscripts ^a^, ^b^ are significantly different, with *p* < 0.05, and ^A^, ^B^ with *p* < 0.01.

**Table 6 animals-14-01285-t006:** Change characteristics of the main microbial communities in the feces of the PRB diet group.

Phylum	Genus	Treatment		SEM	*p*-Value
		CON	PRB		
*Firmicutes*	*Coprococcus*	0.700 ^B^	0.387 ^A^	0.175	0.005
	*Faecalicoccus*	0.018 ^b^	0.085 ^a^	0.005	0.031
*Bacteroidetes*	*Porphyromonas*	0.127 ^a^	0.022 ^b^	0.066	0.049
	*Anaerobacterium*	0.085 ^a^	0.028 ^b^	0.045	0.013

Note: Means in the same row with different superscripts ^a^, ^b^ are significantly different, with *p* < 0.05, and ^A^, ^B^ with *p* < 0.01.

**Table 7 animals-14-01285-t007:** Effect of PRB diet on serum antioxidant capacity of fattening sheep.

Item	Treatment		SEM	*p*-Value
	CON	PRB		
GSH-Px (U/mL)	287.419	288.387	20.213	0.962
MDA (nmol/mL)	2.970	2.179	0.634	0.232
SOD (U/mL)	15.551 ^B^	20.673 ^A^	1.493	0.004
T-AOC (U/mL)	0.288 ^b^	0.473 ^a^	0.082	0.041

Note: Means in the same row with different superscripts ^a^, ^b^ are significantly different, with *p* < 0.05, and ^A^, ^B^ with *p* < 0.01. GSH-Px, glutathione peroxidase; MDA, malondialdehyde; SOD, superoxide dismutase; T-AOC, total antioxidant capacity.

**Table 8 animals-14-01285-t008:** Thirty significantly enriched differential circRNAs in the PRB diet group.

GO.ID ^1^	Term ^2^	Ontology ^3^	Significant ^4^	Annotated ^5^	*p*-Value ^6^
GO:2000758	positive regulation of peptidyl-lysine acetylation	biological process	1/7	23/14,620	0.011
GO:2000756	regulation of peptidyl-lysine acetylation	biological process	1/7	45/14,620	0.021
GO:1990164	histone H2A phosphorylation	biological process	1/7	3/14,620	0.001
GO:1905269	positive regulation of chromatin organization	cellular component	1/7	81/14,620	0.038
GO:1901985	positive regulation of protein acetylation	biological process	1/7	31/14,620	0.015
GO:1901983	regulation of protein acetylation	biological process	1/7	54/14,620	0.026
GO:0090090	negative regulation of canonical Wnt signaling pathway	biological process	1/7	92/14,620	0.043
GO:0071347	cellular response to interleukin-1	biological process	1/7	46/14,620	0.022
GO:0070555	response to interleukin-1	biological process	1/7	56/14,620	0.027
GO:0070498	interleukin-1-mediated signaling pathway	biological process	1/7	17/14,620	0.008
GO:0065009	regulation of molecular function	biological process	3/7	1871/14,620	0.049
GO:0051336	regulation of hydrolase activity	biological process	2/7	806/14,620	0.053
GO:0051092	positive regulation of NF-kappaB transcription factor activity	biological process	1/7	93/14,620	0.043
GO:0043990	histone H2A-S1 phosphorylation	biological process	1/7	1/14,620	0.000
GO:0043988	histone H3-S28 phosphorylation	biological process	1/7	3/14,620	0.001
GO:0043987	histone H3-S10 phosphorylation	biological process	1/7	3/14,620	0.001
GO:0035404	histone–serine phosphorylation	biological process	1/7	7/14,620	0.003
GO:0035066	positive regulation of histone acetylation	biological process	1/7	21/14,620	0.010
GO:0035065	regulation of histone acetylation	biological process	1/7	40/14,620	0.019
GO:0034063	stress granule assembly	biological process	1/7	13/14,620	0.006
GO:0033129	positive regulation of histone phosphorylation	biological process	1/7	5/14,620	0.002
GO:0033127	regulation of histone phosphorylation	biological process	1/7	10/14,620	0.005
GO:0032606	type I interferon production	biological process	1/7	59/14,620	0.028
GO:0031058	positive regulation of histone modification	biological process	1/7	68/14,620	0.032
GO:0031056	regulation of histone modification	biological process	1/7	110/14,620	0.052
GO:0018393	internal peptidyl-lysine acetylation	biological process	1/7	113/14,620	0.053
GO:0016573	histone acetylation	biological process	1/7	111/14,620	0.052
GO:0016572	histone phosphorylation	biological process	1/7	32/14,620	0.015
GO:0006457	protein folding	biological process	1/7	113/14,620	0.053
GO:0002244	hematopoietic progenitor cell differentiation	biological process	1/7	82/14,620	0.039

^1^ GO.ID: number and name of GO. ^2^ Term: description information about GO function. ^3^ Ontology: category of GO (cellular component, biological process, or molecular function). ^4^ Significant: number of differentially expressed circRNA host genes annotated to the GO/number of differentially expressed circRNA host genes annotated to the GO database. ^5^ Annotated: number of genes annotated to the GO/number of genes annotated to the GO database. ^6^
*p*-value: obtained from the *t*-test of the substance in this group comparison.

**Table 9 animals-14-01285-t009:** Gene functions corresponding to 30 significantly enriched differential transcripts in the PRB group.

GO.ID	Term	Ontology	Significant	Annotated	*p*-Value
GO:1903671	negative regulation of sprouting angiogenesis	biological process	1/10	18/14,620	0.012
GO:0090160	Golgi-to-lysosome transport	biological process	1/10	7/14,620	0.005
GO:0090051	negative regulation of cell migration involved in sprouting angiogenesis	biological process	1/10	12/14,620	0.008
GO:0060136	embryonic process involved in female pregnancy	biological process	1/10	5/14,620	0.003
GO:0048757	pigment granule maturation	biological process	1/10	7/14,620	0.005
GO:0045954	positive regulation of natural killer cell-mediated cytotoxicity	biological process	1/10	19/14,620	0.013
GO:0043485	endosome-to-pigment granule transport	biological process	1/10	7/14,620	0.005
GO:0043482	cellular pigment accumulation	biological process	1/10	7/14,620	0.005
GO:0043476	pigment accumulation	biological process	1/10	7/14,620	0.005
GO:0043323	positive regulation of natural killer cell degranulation	biological process	1/10	2/14,620	0.001
GO:0043321	regulation of natural killer cell degranulation	biological process	1/10	2/14,620	0.001
GO:0043320	natural killer cell degranulation	biological process	1/10	6/14,620	0.004
GO:0043302	positive regulation of leukocyte degranulation	biological process	1/10	13/14,620	0.009
GO:0043300	regulation of leukocyte degranulation	biological process	1/10	23/14,620	0.016
GO:0042269	regulation of natural killer cell-mediated cytotoxicity	biological process	1/10	24/14,620	0.016
GO:0035646	endosome-to-melanosome transport	biological process	1/10	7/14,620	0.005
GO:0034058	endosomal vesicle fusion	biological process	1/10	7/14,620	0.005
GO:0032816	positive regulation of natural killer cell activation	biological process	1/10	17/14,620	0.012
GO:0030948	negative regulation of vascular endothelial growth factor receptor signaling pathway	biological process	1/10	8/14,620	0.006
GO:0030947	regulation of vascular endothelial growth factor receptor signaling pathway	biological process	1/10	19/14,620	0.013
GO:0019184	nonribosomal peptide biosynthetic process	biological process	1/10	9/14,620	0.006
GO:0015986	ATP synthesis-coupled proton transport	biological process	1/10	20/14,620	0.014
GO:0015985	energy-coupled proton transport, down electrochemical gradient	biological process	1/10	20/14,620	0.014
GO:0009069	serine family amino acid metabolic process	biological process	1/10	23/14,620	0.016
GO:0006896	Golgi-to-vacuole transport	biological process	1/10	7/14,620	0.005
GO:0006750	glutathione biosynthetic process	biological process	1/10	8/14,620	0.006
GO:0006536	glutamate metabolic process	biological process	1/10	16/14,620	0.011
GO:0006534	cysteine metabolic process	biological process	1/10	10/14,620	0.007
GO:0002717	positive regulation of natural killer cell-mediated immunity	biological process	1/10	19/14,620	0.013
GO:0002323	natural killer cell activation involved in immune response	biological process	1/10	10/14,620	0.007

**Table 10 animals-14-01285-t010:** Gene functions of 30 significantly enriched differential miRNAs targeting mRNA in the PRB group.

GO.ID	Term	Ontology	Gene Ratio ^1^	Bg Ratio ^2^	*p*-Value
GO:0006468	protein phosphorylation	biological process	123/2925	579/27,054	0.000
GO:0045893	positive regulation of transcription, DNA templated	biological process	99/2925	448/27,054	0.000
GO:0045944	positive regulation of transcription from RNA polymerase II promoter	biological process	145/2925	775/27,054	0.000
GO:0007155	cell adhesion	biological process	57/2925	237/27,054	0.000
GO:0006355	regulation of transcription, DNA templated	biological process	173/2925	1051/27,054	0.000
GO:0016310	phosphorylation	biological process	70/2925	326/27,054	0.000
GO:0043547	positive regulation of GTPase activity	biological process	49/2925	197/27,054	0.000
GO:0045892	negative regulation of transcription, DNA templated	biological process	75/2925	360/27,054	0.000
GO:0000122	negative regulation of transcription from RNA polymerase II promoter	biological process	100/2925	533/27,054	0.000
GO:0035556	intracellular signal transduction	biological process	61/2925	279/27,054	0.000
GO:0005654	nucleoplasm	cellular component	399/2925	2246/27,054	0.000
GO:0005634	nucleus	cellular component	521/2925	3186/27,054	0.000
GO:0005829	cytosol	cellular component	385/2925	2176/27,054	0.000
GO:0005737	cytoplasm	cellular component	438/2925	2612/27,054	0.000
GO:0016020	membrane	cellular component	788/2925	5691/27,054	0.000
GO:0005886	plasma membrane	cellular component	381/2925	2418/27,054	0.000
GO:0016021	integral component of membrane	cellular component	709/2925	5188/27,054	0.000
GO:0005887	integral component of plasma membrane	cellular component	91/2925	442/27,054	0.000
GO:0005604	basement membrane	cellular component	24/2925	63/27,054	0.000
GO:0005794	Golgi apparatus	cellular component	122/2925	682/27,054	0.000
GO:0005515	protein binding	molecular function	584/2925	2880/27,054	0.000
GO:0005524	ATP binding	molecular function	262/2925	1328/27,054	0.000
GO:0000166	nucleotide binding	molecular function	193/2925	1042/27,054	0.000
GO:0004672	protein kinase activity	molecular function	115/2925	519/27,054	0.000
GO:0003682	chromatin binding	molecular function	76/2925	287/27,054	0.000
GO:0003677	DNA binding	molecular function	194/2925	1122/27,054	0.000
GO:0046872	metal ion binding	molecular function	234/2925	1462/27,054	0.000
GO:0005509	calcium ion binding	molecular function	110/2925	576/27,054	0.000
GO:0000978	RNA polymerase II core promoter proximal region sequence-specific DNA binding	molecular function	80/2925	380/27,054	0.000
GO:0016301	kinase activity	molecular function	68/2925	319/27,054	0.000

^1^ Gene Ratio: The number of target genes in this GO entry/the number of genes with GO annotations in the target gene. ^2^ Bg Ratio: The number of genes in the GO entry of this entry/the number of genes with GO annotations in all genes.

## Data Availability

The data will be made available on request.

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
