# Peer review of "Multi-Omics Analysis Reveals the Regulatory Mechanism of Probiotics on the Growth Performance of Fattening Sheep"

_animals, 2024, doi:10.3390/ani14091285_

Round 1
Reviewer 1 Report
Comments and Suggestions for Authors
Title Feedback:
The title sounds like a conclusion to the work. "Revealed" would not be the most appropriate term for the title being in the past tense.
Simple summary:
They should dedicate one sentence to refer to the conclusion reached.
Abstract:
Keywords:
Introduction:
L55/56 - you can adapt "and do not have the" to a sentence such as "and presents as advantages" or similar. You should emphasise the benefit of using a probiotic.
L65 - "feed silage" should delete the word "Feed".
L71 - Replace "single stomach animals" with "monogastrics".
L76 - All Latin names should be italicised. Review all scientific names throughout the text.
L93 - This sentence is a little out of context, it should be rewritten and fit into the text.
L99 - "alternative antibiotics" or "probiotics"?
Materials and Methods:
L107 - You should use days or weeks instead of months, avoiding the use of "0.5 months".
Table 1 - what "KIU" means in the legend to table 1.
Figure 1 - The graph is missing links between the text boxes; Is 1.5 × 108 CFU correct? Or is it 1.5 x 108?
They must indicate which test was used to determine the normal distribution of the data. Which T-test was used?
Results:
L204 - The significance value should be indicated since there was no significant increase in the growth of the sheep.
L206 - 208 - This sentence indicated in these lines is not part of the results, it should be removed.
Table 2 - The table legend should indicate what the table refers to, in this case the measurement of rumen tissue thickness. The same goes for table 3, referring to the thickness of the intestine.
L220 - 225 - All the sentences in these lines are not part of the results, so they should be removed.
The representation of the units in tables 2 and 3 is not correct.
In the text, the representation of p-value sometimes appears in lower case (p<) and sometimes in upper case (P<), this should be in accordance with the indications for the author of the journal.
Table 3 legend - it says 3.3, what does that mean?
It should indicate that there are significant differences for which group. You can also choose to use letters instead of *
L239-242 the text should be removed.
Tables 4 and 5 are badly formatted. There is no indication of where there are significant differences.
Tables 4 and 5 are badly formatted. It fails to indicate where there are significant differences.
L243 - 245 - The sentence should be rewritten, it is badly constructed, it also indicates that the ammonia value is higher in the PRB group when compared to the control, then they indicate that there are no differences, in the table we can see that the p value is high (0.140), that is, there is not even a trend, in this parameter although the values are different between the groups, they are similar.
Table 7 - The figure legend lacks an indication of what each parameter means, for example GSH-Px - glutathione peroxidase.
Discussion:
L477 -478 - What results do you have to back up this statement?
L484 - 486 - How can you say that the results of your research can improve the growth of poultry? This work was only carried out with sheep.
L495 - You should only refer to ruminants instead of ruminant stomachs.
The statement in L495 lacks a bibliographical reference.
Every time they use expressions such as significant, statistically significant, significant differences, non-significant or similar expressions, they should always be accompanied by a p-value.
Throughout the discussion there are various statements that are not referenced and that do not come from the results of the study. All these statements should be accompanied by the author.
Conclusion:
The last two sentences are the conclusion of the paper.
References:
The references in the text and at the end are not in accordance with the instructions for authors. During the text they should appear in the form of a number [1], [2], etc... and at the end they should follow this order.
The entire text needs to be re-formatted.
Author Response
Responses to reviewers’ comments
Manuscript ID: animals-2954007
Title: Multi-omics analysis reveals the regulatory mechanism of probiotics on the growth performance of fattening sheep.
Dear Editor and Reviewers:
We are truly grateful to you and the reviewers for the critical comments and thoughtful suggestions on our manuscript. They are really helpful and based on these comments and suggestions, we have revised the manuscript carefully. In the following pages are our point-by-point responses to the reviewers’ comments/suggestions. Please feel free to contact us if there is any question and we are very willing to improve our manuscript until all the reviewers are satisfied.
Best regards,
Hongguo Cao
Address: College of Animal Science and Technology, Anhui Agricultural University, Hefei 230036, P.R. China
E-mail: caohongguo1@ahau.edu.cn
Reviewer:1
Comment: Title Feedback: The title sounds like a conclusion to the work. "Revealed" would not be the most appropriate term for the title being in the past tense.
Answer: Thank you for pointing out the problem. According to your comments and suggestions, we have made modifications to the title. (Line 2-3)
C: Simple summary: They should dedicate one sentence to refer to the conclusion reached.
A: Thank you for pointing out the problem. According to your comments and suggestions, we summarized the conclusion in one sentence. (Line16-17)
C: L55/56 - you can adapt "and do not have the" to a sentence such as "and presents as advantages" or similar. You should emphasise the benefit of using a probiotic.
A: Thank you for pointing out the problem. According to your comments and suggestions, we emphasized the benefits of using probiotics and rewrote the sentences. (Line 53)
C: L65 - "feed silage" should delete the word "Feed".
A: Thank you for pointing out the problem. According to your comments and suggestions, we have removed the word "Feed". (Line 61)
C: L71 - Replace "single stomach animals" with "monogastrics".
A: Thank you for pointing out the problem. We are very sorry for our carelessness. We replaced "single stomach animals" with "monogastrics". (Line 66)
C: L76 - All Latin names should be italicised. Review all scientific names throughout the text.
A: Thank you for pointing out the problem. According to your suggestion, we used italics for all Latin names throughout the article. (Line 31-543)
C: L93 - This sentence is a little out of context, it should be rewritten and fit into the text.
A: Thank you for pointing out the problem. According to your suggestion, we have rewritten this sentence. (Line 85)
C: L99 - "alternative antibiotics" or "probiotics"?
A: Thank you for pointing out the problem. According to your suggestion, we believe it is probiotics because excessive abuse of antibiotics can cause bacteria in animals to develop resistance and disrupt the balance of the animal gastrointestinal microbiota. The purpose of our experiment is to reduce the use of antibiotics through probiotics. (Line 90)
C: L107 - You should use days or weeks instead of months, avoiding the use of "0.5 months".
A: Thank you for pointing out the problem. According to your comments and suggestions, we have changed to 2 weeks. (Line 99)
C: Table 1 - what "KIU" means in the legend to table 1.
A: Thank you for pointing out the problem. According to your comments and suggestions, we have changed to 600 000 IU of Vitamin A, 80 000 IU of vitamin D3. (Line 106)
C: Figure 1 - The graph is missing links between the text boxes; Is 1.5 × 108 CFU correct? Or is it 1.5 × 108 ?
A: Thank you for pointing out the problem. According to your comments and suggestions, we have modified the graph by changing the 1.5× 108 CFU to 1.5× 108 CFU/g. (Line 111)
C: They must indicate which test was used to determine the normal distribution of the data. Which T-test was used?
A: Thank you for pointing out the problem. We are very sorry for our carelessness, we used the SPSSAU data analysis platform to perform a normal distribution test on the data. After confirming that the data conforms to the normal distribution, we conducted independent sample t-tests on the data. (Line 180)
C: L204 - The significance value should be indicated since there was no significant increase in the growth of the sheep.
A: Thank you for pointing out the problem. We are very sorry for our carelessness, we have indicated the significance value (p>0.05). (Line 192)
C: L206 - 208 - This sentence indicated in these lines is not part of the results, it should be removed.
A: Thank you for pointing out the problem. According to your comments and suggestions, we have deleted this sentence. (Line 195)
C: Table 2 - The table legend should indicate what the table refers to, in this case the measurement of rumen tissue thickness. The same goes for table 3, referring to the thickness of the intestine. The representation of the units in tables 2 and 3 is not correct.
A: Thank you for pointing out the problem. According to your comments and suggestions, we have made modifications to the legends and units in Tables 2 and 3. (Line 208 and 230)
C: L220 - 225 - All the sentences in these lines are not part of the results, so they should be removed. A: Thank you for pointing out the problem. According to your comments and suggestions, we have deleted these sentences. (Line 215)
C: In the text, the representation of p-value sometimes appears in lower case (p<) and sometimes in upper case (P<), this should be in accordance with the indications for the author of the journal.
A: Thank you for pointing out the problem. According to your comments and suggestions, we have standardized the p-value throughout the entire text in lowercase. (Line 11-587)
C: Table 3 legend - it says 3.3, what does that mean?
A: Thank you for pointing out the problem. We are very sorry for our carelessness, we have modified the table legends. (Line 233)
C: It should indicate that there are significant differences for which group. You can also choose to use letters instead of *
A: Thank you for pointing out the problem. According to your comments and suggestions, we replaced * with letters. (Line 230)
C: L239-242 the text should be removed.
A: Thank you for pointing out the problem. According to your comments and suggestions, we have deleted this paragraph. (Line 235-238)
C: Tables 4 and 5 are badly formatted. There is no indication of where there are significant differences. Tables 4 and 5 are badly formatted. It fails to indicate where there are significant differences.
A: Thank you for pointing out the problem. According to your comments and suggestions, we have modified Tables 4 and 5. (Line 247-262)
C: L243 - 245 - The sentence should be rewritten, it is badly constructed, it also indicates that the ammonia value is higher in the PRB group when compared to the control, then they indicate that there are no differences, in the table we can see that the p value is high (0.140), that is, there is not even a trend, in this parameter although the values are different between the groups, they are similar.
A: Thank you for pointing out the problem. According to your comments and suggestions, we have modified the sentence. (Line 240)
C: Table 7 - The figure legend lacks an indication of what each parameter means, for example GSH-Px - glutathione peroxidase.
A: Thank you for pointing out the problem. We are very sorry for our carelessness. we have added an indication of the meaning of each parameter. (Line 326)
C: L477 -478 - What results do you have to back up this statement?
A: Thank you for pointing out the problem. According to your suggestion, we cited relevant references to support this statement. (Line 482)
C: L484 - 486 - How can you say that the results of your research can improve the growth of poultry? This work was only carried out with sheep.
A: Thank you for pointing out the problem. According to your suggestion, we have revised this sentence. (Line 489)
C: L495 - You should only refer to ruminants instead of ruminant stomachs. The statement in L495 lacks a bibliographical reference.
A: Thank you for pointing out the problem. According to your comments and suggestions, we have rewritten this sentence and added references. (Line 497)
C: Every time they use expressions such as significant, statistically significant, significant differences, non-significant or similar expressions, they should always be accompanied by a p-value.
A: Thank you for pointing out the problem. According to your comments and suggestions, we have reviewed the entire text and always included p-values when using expressions such as significance, statistical significance, significant differences, non significance, or similar expressions. (Line 19-601)
C: Throughout the discussion there are various statements that are not referenced and that do not come from the results of the study. All these statements should be accompanied by the author.
A: Thank you for pointing out the problem. According to your comments and suggestions, we have reviewed and revised the entire discussion, adding relevant references. (Line 481-587)
C: Conclusion: The last two sentences are the conclusion of the paper.
A: Thank you for pointing out the problem. According to your comments and suggestions, we have revised the conclusion of the paper. (Line 608-613)
C: References: The references in the text and at the end are not in accordance with the instructions for authors. During the text they should appear in the form of a number [1], [2], etc... and at the end they should follow this order.
A: Thank you for pointing out the problem. According to your comments and suggestions, we have corrected the relevant references throughout the text. (Line 632-770)
We tried our best to revise manuscript. These changes will not influence the content and framework of the manuscript. We appreciate for editor’s and reviewers’ critical comments and thoughtful suggestions for our manuscript and hope that the revised manuscript will meet the standard of Animals.
Once again, thank you very much for your comments and suggestions.
Sincerely Yours,
Hongguo Cao
Reviewer 2 Report
Comments and Suggestions for Authors
Multi-omics Analysis Revealed the Regulatory Mechanism of Probiotics on the Growth Performance of Fattening Sheep
Dear Authors,
The manuscript is very interesting, and describes important issue which is effect of probiotics on the performance of fattening sheep with the regulatory mechanisms of its action. But there are some aspects to correct. Mainly description of data analysis (normality test of data distribution, F-test and equality of variances). That will be important because number of animals is low (12 sheep), and there is no information about its variability. Beside of this some corrections are required in case of tables and description of significance level between treatments.
Below I add some suggestions helpful in this process:
Line 2
Multi-omics without capitalization could be used.
Line 22
12 sheep in experiment gives very low power of a test: 6 in each treatment, the origin of all animals must be very close in this case. In case of higher variability better is to have more observations, even replications with 3-5 animals inside.
Line 24
In text of manuscript is: 1.5 x 108 CFU, but the quantity must be expressed in unit of surface, volume, mass, perhaps in this case it is 1.5 x 108 CFU/g.
Line 26-533
P-value expressed as (P<0.05), must be expressed as (p<0.05)
Line 32
To present name of genus, italics are required.
Line 50
In Animals as the same like in other MDPI Journals references are presented in numbers (sometimes is controversial because Authors doesn’t want to be treated as numbers, but it helps during review process) , please check Instructions for Authors, but in this case it is not complicated [1,2] in case line 50, in case of line 57 are more than 2 authors in order [3-5], and so on…
Line 76-603
In case of genus name or in binominal name of species italics must be used.
Line 115
Table 1
% of crude protein in soybean meal and rapeseed meal will be useful information.
Last ingredient, Probiotics, unit per g probably must be used.
Are you sure that samples in 3 repetitions for two diets (without and with probiotic) gives the same results in case of chemical composition ?
Phosphorus (total or available?).
Metabolizable energy must be expressed in MJ/kg.
Line 116
In case of genus name or in binominal name of species italics must be used.
Line 124
In text is ‘1.5 x 108 CFU’, must be 1.5 x 108 CFU/g.
Line 126 and 131
In text is 1st and 60th, ordinal numeral ending must be in superscript: 1st and 60th.
Line 178 and 182
p-value
Line 186
Tool for normal distribution verification of data is described (in SPSS), but in this case name of test will be also required, probably Shapiro-Wilk’s test. In case low power of a test, F-test of equality of variances also will gives information about variability of animals in treatments.
2.6. Data Analysis subsection must be rewrote. 2 times using is used in 2 following sentences (passive voice can be used, ie.:
To organize the data Excel 2016 was used (passive voice, past simple).
Information about SEM required. In Table 2 and 3 there are presented means and sd, information about it is also required or data can be expressed the same like in table 4
In the last sentence in subsection 2.6 can be added information: The statistical significant differences was determined at two levels corresponding with p<0.05 and p<0.01.
Line 215
Minimally information about p-value in one more added column required.
Line 236
Same like in line 215.
Line 237
Subsection 3.3 must be transferred to line 239.
Line 252
Table 4
p-value in table
|
Total volatile fatty acid , mM |
66.10B |
88.69A |
4.055 |
0.000 |
|
Acetate, mM |
45.23B |
61.48A |
3.258 |
0.000 |
|
Propionate, mM |
12.21B |
18.07A |
1.090 |
0.000 |
Under the table information can be added: Means in the same row with different superscripts A,B are significantly different with p<0.01.
Line 265
Table 5
Name of different genus of bacteria can be present in italics form, except Subdivision5_genera_incertae_sedis.
Under the table information can be added: Means in the same row with different superscripts a,b are significantly different with p<0.05 and A,B with p<0.01.
Line 277
Table 6
The same as in line 265.
Line 322
Table 7
The same as in line 265.
Line 408, 443 and 472
Table 8, 9 and 10
p-value instead of P-value.
Line 628-729
References must be also adapted to MDPI pattern (Instructions for Authors).
Abbreviations with dots, Journal name with italics, year of publication with bold and doi on the end of reference.
Ie. No.1:
Abd El-Hack, M.E.; El-Saadony, M.T.; Shafi, M.E.; Qattan, S.Y.A.; Batiha, G.E.; Khafaga, A.F.; Abdel-Moneim, A.E.; Alagawany, M. Probiotics in poultry feed: A comprehensive review. J. Anim. Physiol. Anim. Nutr. (Berl.) 2020, 104(6):1835-1850. https://doi.org/10.1111/jpn.13454
Author Response
Responses to reviewers’ comments
Manuscript ID: animals-2954007
Title: Multiomics analysis reveals the regulatory mechanism of probiotics on the growth performance of fattening sheep
Dear Editor and Reviewers:
We are truly grateful to you and the reviewers for the critical comments and thoughtful suggestions on our manuscript. They are really helpful and based on these comments and suggestions, we have revised the manuscript carefully. In the following pages are our point-by-point responses to the reviewers’ comments/suggestions. Please feel free to contact us if there is any question and we are very willing to improve our manuscript until all the reviewers are satisfied.
Best regards,
Hongguo Cao
Address: College of Animal Science and Technology, Anhui Agricultural University, Hefei 230036, P.R. China
E-mail: caohongguo1@ahau.edu.cn
Reviewer:2
Comment: Line 2-Multi-omics without capitalization could be used.
Answer: Thank you for pointing out the problem. According to your comments and suggestions, we have made modifications to the title. (Line2- Line3)
C: Line 22-12 sheep in experiment gives very low power of a test: 6 in each treatment, the origin of all animals must be very close in this case. In case of higher variability better is to have more observations, even replications with 3-5 animals inside.
A: Thank you for pointing out the problem. Because animal experimental design should follow the "3R principle", including the principles of animal substitution, reduction, and optimization. In statistics, it is usually required that each group has at least 6 available data to be meaningful. In the experimental design, the experimental group and the control group each had 6 sheep to reduce economic losses. In the future, we will conduct in-depth research on the mechanism through a larger sample size and promote the application of research results in healthy aquaculture. (Line 22)
C: Line 24-In text of manuscript is: 1.5 x 108 CFU, but the quantity must be expressed in unit of surface, volume, mass, perhaps in this case it is 1.5 x 108 CFU/g.
A: Thank you for pointing out the problem. According to your comments and suggestions, we have changed the unit to 1.5 x 108 CFU/g. (Line 24)
C: Line 26-533-P-value expressed as (P<0.05), must be expressed as (p<0.05)
A: Thank you for pointing out the problem. According to your comments and suggestions, we have changed the (P<0.05) throughout the text to (p<0.05). (Line 26-533)
C: In Animals as the same like in other MDPI Journals references are presented in numbers (sometimes is controversial because Authors doesn’t want to be treated as numbers, but it helps during review process) , please check Instructions for Authors, but in this case it is not complicated [1,2] in case line 50, in case of line 57 are more than 2 authors in order [3-5], and so on…
A: Thank you for pointing out the problem. According to your suggestion, we have modified the reference format. (Line 632-770)
C: Line 32-To present name of genus, italics are required.
Line 76-603-In case of genus name or in binominal name of species italics must be used.
A: Thank you for pointing out the problem. According to your suggestion, we have applied italics to the binomial names of genera or species throughout the entire article. (Line 69-603)
C: Line 115-Table 1 % of crude protein in soybean meal and rapeseed meal will be useful information. Last ingredient, Probiotics, unit per g probably must be used. Metabolizable energy must be expressed in MJ/kg.
A: Thank you for pointing out the problem. According to your suggestion, we added the percentage of crude protein in soybean meal and rapeseed meal, and modified the units CFU/g and MJ/kg according to the suggestions. (Line 106)
C: Are you sure that samples in 3 repetitions for two diets (without and with probiotic) gives the same results in case of chemical composition ? Phosphorus (total or available?).
A: Thank you for pointing out the problem. we obtained the results based on the average of three repeated samples of two diets (without probiotics and with probiotics); The phosphorus in the table is total phosphorus. (Line 106)
C: Line 116-In case of genus name or in binominal name of species italics must be used. Line 124-In text is ‘1.5 x 108 CFU’, must be 1.5 x 108 CFU/g.
A: Thank you for pointing out the problem. According to your comments and suggestions, we have applied italics to the binomial names of genera or species, and changed the unit to 1.5 x 108 CFU/g. (Line 115)
C: Line 126 and 131-In text is 1st and 60th, ordinal numeral ending must be in superscript: 1st and 60th.
A: Thank you for pointing out the problem. According to your comments and suggestions, we have superscripted the end of the number. (Line 117 and 122)
C: Line 178 and 182-p-value
A: Thank you for pointing out the problem. We are very sorry for our carelessness, we have modified Pvalue to p-value. (Line 168-172)
C: Tool for normal distribution verification of data is described (in SPSS), but in this case name of test will be also required, probably Shapiro-Wilk’s test. In case low power of a test, F-test of equality of variances also will gives information about variability of animals in treatments.
A: Thank you for pointing out the problem. We are very sorry for our carelessness, we used the SPSSAU data analysis platform to perform a normal distribution test on the data. The Shapiro-Wilk test was used for normality testing. After confirming that the data conforms to the normal distribution, we conducted independent sample t-tests on the data. (Line 178-182)
C: 2.6. Data Analysis subsection must be rewrote. 2 times using is used in 2 following sentences (passive voice can be used, ie.: To organize the data Excel 2016 was used (passive voice, past simple).
A: Thank you for pointing out the problem. According to your comments and suggestions, we have rewritten the data analysis section. (Line 179)
C: Information about SEM required. In Table 2 and 3 there are presented means and sd, information about it is also required or data can be expressed the same like in table 4
A: Thank you for pointing out the problem. According to your comments and suggestions, we have made modifications to Tables 2 and 3. (Line 208 and 232)
C: In the last sentence in subsection 2.6 can be added information: The statistical significant differences was determined at two levels corresponding with p<0.05 and p<0.01.
A: Thank you for pointing out the problem. According to your comments and suggestions, we added this sentence. (Line 182)
C: Line 215-Minimally information about p-value in one more added column required. Line 236-Same like in line 215.
A: Thank you for pointing out the problem. According to your comments and suggestions, we have reorganized the table. (Line 208 and 233)
C: Line 237-Subsection 3.3 must be transferred to line 239.
A: Thank you for pointing out the problem. We are very sorry for our carelessness. We moved section 3.3. (Line 233)
C: Line 252-Table 4 p-value in table. Under the table information can be added: Means in the same row with different superscripts A, B are significantly different with p<0.01.
A: Thank you for pointing out the problem. According to your comments and suggestions, we have modified Table 4. (Line 248)
C: Line 265、277、322 Table 5、Table 6 、Table 7
Name of different genus of bacteria can be present in italics form, except Subdivision5_genera_incertae_sedis. Under the table information can be added: Means in the same row with different superscripts a, b are significantly different with p<0.05 and A,B with p<0.01.
A: Thank you for pointing out the problem. According to your comments and suggestions, we have modified Tables 5, 6, and 7. (Line 262-327)
C: Line 408, 443 and 472 Table 8, 9 and 10 p-value instead of P-value
A: Thank you for pointing out the problem. According to your comments and suggestions, we have changed the P-value in Tables 8, 9, and 10 to p-value. (Line 412-475)
C: Line 628-729
References must be also adapted to MDPI pattern (Instructions for Authors).
Abbreviations with dots, Journal name with italics, year of publication with bold and doi on the end of reference.
A: Thank you for pointing out the problem. According to your comments and suggestions, we have corrected the relevant references throughout the text. (Line 632-770)
We tried our best to revise manuscript. These changes will not influence the content and framework of the manuscript. We appreciate for editor’s and reviewers’ critical comments and thoughtful suggestions for our manuscript and hope that the revised manuscript will meet the standard of Animals.
Once again, thank you very much for your comments and suggestions.
Sincerely Yours,
Hongguo Cao
Reviewer 3 Report
Comments and Suggestions for Authors
Dear Authors
Thank you for undertaking this body of work.
Generally, I found the manuscript to be well written and relatively easy to comprehend. I liked the effort made to combine multiple measurement/analysis in order to obtain result for several important measures.
The Introduction provided an adequate background to the topic.
The Materials and Methods were mostly adequate in the descriptions given but you must revise this section to include the details of the species of bacteria in the probiotic provided to the experimental sheep. The statistical analysis was appropriate for the data obtained in this study.
The Results were adequately written but I would suggest the text should be focussed on the important results from the tables rather than providing a description of the findings in the tables (as that is a little repetitive).
The Discussion was generally well written but the final paragraph needs to be revised. The same can be said of the Conclusion. I will comment on the suggested revisions below.
Specific Comments -
The footnoting in the list of authors is messy (different font sizes and spaces in some cases but not in others). This needs re-formatting.
At line 106 insert the full details of the probiotic used in the study.
At line 129 the calculation was made to arrive at "average daily feed intake" not "gain".
Line 137 - should read "and store in liquid nitrogen for later use."
Line 158 - Analyzed should have a lower case 'a'.
Line 178 and 182 etc - re-format the Pvalue< to have spaces between the P and value etc.
Lines 186 and 187 - this sentence doesn't make sense - re-write it please.
Line 203 mentions a "tetravalent" probiotic - consistency in the description needs to be adhered to.
Line 250 - it is simply the concentration of butyric acid and valeric acid, and do you mean the acetic acid:propionic acid ratio? Why use "butyric acid" in the text and "Butyrate" in the table (even though they are the same substance)? That may confuse some readers. The VFAs should be referred to in a consistent manner.
Line 254 - It is Prevotella (not Prevolella) - this occurs elsewhere too.
Line 256 - Prevotella please, and Methanobrevibacter (not Metanobrevibater).
Line 274 - use the upper case 'F' in faecalicoccus.
Lines 298/299 separate the words "metabolism were"
Line 372 - the citation should be Yang et al., 2022
Line 373 - citation should be Zhou et al., 2020
Line 412/413 - there is no description provided for footnote 6 "P value"
Line 449 - citations only need the first author then et al.,
Line 471 - Table 10 - Is it possible that all P values are 0.000?
Lines 580 - 585 - this is one very long sentence that loses sense and the citations need to be fixed. I strongly recommend this sentence be broken down into several simpler single topic sentences (and don't use etc.).
Lines 585 - 590 - Re-write this sentence. "proving" is not the best choice of word - perhaps you should consider using "supporting the finding" or similar. It is possible that any one or more in combination of the mechanisms you have mentioned could be responsible for the enhanced growth. Your conclusion is a little too deliberate.
Line 611 onwards to 616. - I would not say "fully demonstrate here. I think you might like to say the "probiotics alter the GI microbiota ..." "and this resulted in improved growth performance ...". This part of the conclusion needs revision.
The References appear to be adequate and appropriate but I have not performed a cross-check for accuracy.
Thank you again for undertaking the work.
Comments on the Quality of English LanguageSee "Comments to Authors"
Author Response
Responses to reviewers’ comments
Manuscript ID: animals-2954007
Title: Multi-omics analysis reveals the regulatory mechanism of probiotics on the growth performance of fattening sheep.
Dear Editor and Reviewers:
We are truly grateful to you and the reviewers for the critical comments and thoughtful suggestions on our manuscript. They are really helpful and based on these comments and suggestions, we have revised the manuscript carefully. In the following pages are our point-by-point responses to the reviewers’ comments/suggestions. Please feel free to contact us if there is any question and we are very willing to improve our manuscript until all the reviewers are satisfied.
Best regards,
Hongguo Cao
Address: College of Animal Science and Technology, Anhui Agricultural University, Hefei 230036, P.R. China
E-mail: caohongguo1@ahau.edu.cn
Reviewer:3
Comment: The footnoting in the list of authors is messy (different font sizes and spaces in some cases but not in others). This needs re-formatting.
Answer: Thank you for pointing out the problem. According to your comments and suggestions, we have made modifications to the footnotes in the author list. (Line 4)
C: At line 106 insert the full details of the probiotic used in the study.
A: Thank you for pointing out the problem. According to your comments and suggestions, we have added all detailed information on the probiotics used. (Line 98)
C: At line 129 the calculation was made to arrive at "average daily feed intake" not "gain".
A: Thank you for pointing out the problem. According to your comments and suggestions, we have revised this sentence. (Line 120)
C: Line 137 - should read "and store in liquid nitrogen for later use."
A: Thank you for pointing out the problem. According to your comments and suggestions, we have revised this sentence. (Line 128)
C: Line 158 - Analyzed should have a lower case 'a'.
A: Thank you for pointing out the problem. We are very sorry for our carelessness. We replaced "A" with "a". (Line 148)
C: Line 178 and 182 etc - re-format the Pvalue< to have spaces between the P and value etc.
A: Thank you for pointing out the problem. According to your suggestion, we changed Pvalue to p-value. (Line 168-173)
C: Lines 186 and 187 - this sentence doesn't make sense - re-write it please.
A: Thank you for pointing out the problem. According to your suggestion, we have rewritten this sentence. (Line 178-182)
C: Line 203 mentions a "tetravalent" probiotic - consistency in the description needs to be adhered to.
A: Thank you for pointing out the problem. According to your suggestion, we have deleted this sentence. (Line 195)
C: Line 250 - it is simply the concentration of butyric acid and valeric acid, and do you mean the acetic acid: propionic acid ratio? Why use "butyric acid" in the text and "Butyrate" in the table (even though they are the same substance)? That may confuse some readers. The VFAs should be referred to in a consistent manner.
A: Thank you for pointing out the problem. We are very sorry for our carelessness, we have revised this paragraph. (Line 244-246)
C: Line 254 - It is Prevotella (not Prevolella) - this occurs elsewhere too. Line 256 - Prevotella please, and Methanobrevibacter (not Metanobrevibater).
A: Thank you for pointing out the problem. We are very sorry for our carelessness, we have changed "Prevotella" to "Prevotella" and "Metanobrevibacter" to "Metanobrevibacter" throughout the text. (Line 251 and 258)
C: Line 274 - use the upper case 'F' in faecalicoccus. Lines 298/299 separate the words "metabolism were"
A: Thank you for pointing out the problem. We are very sorry for our carelessness, we changed 'f' to 'F' and separated 'metabolismwere' with spaces. (Line 258 and 273)
C: Line 372 - the citation should be Yang et al., 2022
Line 373 - citation should be Zhou et al., 2020
Line 449 - citations only need the first author then et al.,
A: Thank you for pointing out the problem. According to your comments and suggestions, we have corrected the relevant references throughout the text. (Line 632-770)
C: Line 412/413 - there is no description provided for footnote 6 "P value"
A: Thank you for pointing out the problem. We are very sorry for our carelessness, we have provided an explanation for "p-value". (Line 417)
C: Line 471 - Table 10 - Is it possible that all P values are 0.000?
A: Thank you for pointing out the problem. According to your comments and suggestions, our experimental results show that all p values in Table 10 are 0.000. (Line 476)
C: Lines 580 - 585 - this is one very long sentence that loses sense and the citations need to be fixed. I strongly recommend this sentence be broken down into several simpler single topic sentences (and don't use etc.).
A: Thank you for pointing out the problem. According to your comments and suggestions, we have made modifications to this paragraph. (Line 580-582)
C: Lines 585 - 590 - Re-write this sentence. "proving" is not the best choice of word - perhaps you should consider using "supporting the finding" or similar. It is possible that any one or more in combination of the mechanisms you have mentioned could be responsible for the enhanced growth. Your conclusion is a little too deliberate.
A: Thank you for pointing out the problem. According to your comments and suggestions, we have rewritten this sentence. (Line 582-587)
C: Line 611 onwards to 616. - I would not say "fully demonstrate here. I think you might like to say the "probiotics alter the GI microbiota ..." "and this resulted in improved growth performance ...". This part of the conclusion needs revision.
A: Thank you for pointing out the problem. According to your comments and suggestions, we have made modifications to the conclusion. (Line 585-613)
We tried our best to revise manuscript. These changes will not influence the content and framework of the manuscript. We appreciate for editor’s and reviewers’ critical comments and thoughtful suggestions for our manuscript and hope that the revised manuscript will meet the standard of Animals.
Once again, thank you very much for your comments and suggestions.
Sincerely Yours,
Hongguo Cao
Round 2
Reviewer 1 Report
Comments and Suggestions for Authors
the authors introduced all the recommended changes